# On the Impact of Hard Adversarial Instances on Overfitting in Adversarial Training

## Abstract

Adversarial training is a popular method to robustify models against adversarial attacks. However, it exhibits much more severe overfitting than training on clean inputs. In this work, we investigate this phenomenon from the perspective of training instances, i.e., training input-target pairs. We provide a quantitative metric measuring the difficulty of an instance in the training set and analyze the model's behavior on instances of different difficulty levels. This lets us show that the decay in generalization performance of adversarial training is a result of the model's attempt to fit hard adversarial instances. We theoretically verify our observations for both linear and general nonlinear models, proving that models trained on hard instances have worse generalization performance than ones trained on easy instances, and that this generalization gap is larger in adversarial training. Finally, we investigate solutions to mitigate adversarial overfitting in several scenarios, including when relying on fast adversarial training and in the context of fine-tuning a pretrained model with additional data. Our results demonstrate that using training data adaptively improves the model's robustness.

## 1 Introduction

The existence of adversarial examples (Szegedy et al., 2014) causes serious safety concerns when deploying modern deep learning models. For example, for classification tasks, imperceptible perturbations of the input instance can fool state-of-the-art classifiers. Many strategies to obtain models that are robust against adversarial attacks have been proposed (Buckman et al., 2018; Dhillon et al., 2018; Ma et al., 2018; Pang et al., 2020; 2019; Samangouei et al., 2018; Xiao et al., 2020), but most of them have been found to be ineffective in the presence of adaptive attacks (Athalye et al., 2018; Croce & Hein, 2020b; Tramer et al., 2020). Ultimately, this leaves adversarial training (Madry et al., 2018) and its variants (Alayrac et al., 2019; Carmon et al., 2019; Gowal et al., 2020; Hendrycks et al., 2019; Kumari et al., 2019; Wu et al., 2020; Zhang et al., 2019a) as the most effective and popular approach to construct robust models. Unfortunately, adversarial training yields much worse performance on the test data than vanilla training. In particular, it strongly suffers from overfitting (Rice et al., 2020), with the model's performance decaying significantly on the test set in the later phase of adversarial training. While this can be mitigated by early stopping (Rice et al., 2020) or model smoothing (Chen et al., 2021b), the reason behind the overfitting of adversarial training remains poorly understood.

In this paper, we study this phenomenon from the perspective of training instances, i.e., training input-target pairs. We introduce a quantitative metric to measure the difficulty of the instances and analyze the model's behavior, such as its loss and intermediate activations, on training instances of different difficulty levels. This lets us discover that the model's generalization performance decays significantly when it fits the hard adversarial instances in the later training phase.

To more rigorously study this phenomenon, we then perform theoretical analyses on both linear and nonlinear models. For linear models, we study logistic regression on a Gaussian mixture model, in which we can calculate the analytical expression of the model parameters upon convergence and thus the robust test accuracy. Our theorem demonstrates that adversarial training on harder instances leads to larger generalization gaps. We further prove that the gap in robust test accuracy between models trained by hard instances and ones trained by easy instances increases with the size of the adversarial budget. In the case of nonlinear models, we derive the lower bound of the model's Lipschitz constant when the model is well fit to the adversarial training examples. This bound increases with the

difficulty level of the training instances and the size of the adversarial budget. Since a larger Lipschitz constant indicates a higher adversarial vulnerability (Ruan et al., 2018; Weng et al., 2018a;b), our theoretical analysis confirms our empirical observations.

Our findings can be broadly applied to obtain robust models. To evidence this, in addition to the standard adversarial training settings of (Madry et al., 2018), we study the following two scenarios: fast adversarial training and adversarial fine-tuning with additional training data. Our proposed method assigns adaptive targets or adaptive weights on training instances to avoid fitting hard input-target pairs. We show it can mitigate adversarial overfitting and improve models' performance. For fast adversarial training, our results are better than other accelerated adversarial training methods available on RobustBench (Croce et al., 2020). For adversarial fine-tuning with additional training data, we show improved performance over the methods in (Alayrac et al., 2019; Carmon et al., 2019).

**Contributions.** In summary, our contributions are as follows: 1) Based on a quantitative metric of instance difficulty, we show that fitting hard adversarial instances leads to degraded generalization performance in adversarial training. 2) We conduct a rigorous theoretical analysis on both linear and nonlinear models. For linear models, we show analytically that models trained on harder instances have larger robust test error than the ones trained on easy instances; the gap increases with the size of the adversarial budget. For nonlinear models, we derive a lower bound of the model's Lipschitz constant. The lower bound increases with the difficulty of the training instances and the size of the adversarial budget, indicating both factors make adversarial overfitting more severe. 3) We show that the adaptive use of the easy and hard training instances can improve the performance in fast adversarial training and adversarial fine-tuning with additional training data.

**Notation and terminology.** In this paper, $x$ and $x'$ are the clean input and its adversarial counterpart. We use $f_w$ to represent a model parameterized by $w$ and omit the subscript $w$ unless ambiguous. $o = f_w(x)$ and $o' = f_w(x')$ are the model's output of the clean input and the adversarial input. $\mathcal{L}_w(x, y)$ and $\mathcal{L}_w(x', y)$ represent the loss of the clean and adversarial instances, receptively, in which we sometimes omit $w$ and $y$ for notation simplicity. We use $\|w\|$ and $\|\mathbf{X}\|$ to represent the $l_2$ norm of the vector $w$ and the largest singular value of the matrix $\mathbf{X}$, respectively. $sign$ is an elementwise function which returns $+1$ for positive elements, $-1$ for negative elements and $0$ for $0$. $\mathbf{1}_y$ is the one-hot vector with only the $y$-th dimension being $1$. The term *adversarial budget* refers to the allowable perturbations applied to the input instance. It is characterized by $l_p$ norm and the size $\epsilon$ as a set $\mathcal{S}^{(p)}(\epsilon) = \{\Delta | \|\Delta\|_p \leq \epsilon\}$, with $\epsilon$ defining the budget size. Therefore, given the training set $\mathcal{D}$, the robust learning problem can be formulated as the min-max optimization $\min_w \mathbb{E}_{x \sim \mathcal{D}} \max_{\Delta \in \mathcal{S}^{(p)}(\epsilon)} \mathcal{L}_w(x + \Delta)$. A notation table is provided in Appendix A.

In this paper, *vanilla training* refers to training on the clean inputs, and *vanilla adversarial training* to the adversarial training method in (Madry et al., 2018). *RN18* and *WRN34* are the 18-layer ResNet (He et al., 2016) and the 34-layer WideResNet (Zagoruyko & Komodakis, 2016) used in (Madry et al., 2018) and (Wong et al., 2020), respectively. To avoid confusion with the general term *overfitting*, which denotes the gap between the training and test accuracy, we employ the term *adversarial overfitting* to indicate the phenomenon that robust accuracy on the test set decreases significantly in the later phase of vanilla adversarial training. This phenomenon was pointed out in (Rice et al., 2020) and does not occur in vanilla training. Our code is submitted on GoogleDrive anonymously[1].

## 2 RELATED WORK

We concentrate on white-box attacks, where the attacker has access to the model parameters. Such attacks are usually based on first-order information and stronger than black-box attacks (Andriushchenko et al., 2020; Dong et al., 2018). For example, the *fast gradient sign method (FGSM)* (Goodfellow et al., 2014) perturbs the input based on its gradient's sign, i.e., $\Delta = \epsilon \, sign(\nabla_x \mathcal{L}_w(x))$. The *iterative fast gradient sign method (IFGSM)* (Kurakin et al., 2016) iteratively runs FGSM using a smaller step size and projects the perturbation back to the adversarial budget after each iteration. On top of IFGSM, *projected gradient descent (PGD)* (Madry et al., 2018) use random initial perturbations and restarts to boost the strength of the attack.

Many methods have been proposed to defend a model against adversarial attacks (Buckman et al., 2018; Dhillon et al., 2018; Ma et al., 2018; Pang et al., 2020; 2019; Samangouei et al., 2018; Xiao

---

[1]https://drive.google.com/file/d/1vb6ehNMkBeNIM3dLr_igKMUcCgBd9ZmK/view?usp=sharing

et al., 2020). However, most of them were shown to utilize obfuscated gradients (Athalye et al., 2018; Croce & Hein, 2020b; Tramer et al., 2020), that is, training the model to tackle some specific types of attacks instead of achieving true robustness. This makes these falsely robust models vulnerable to stronger adaptive attacks. By contrast, several works have designed training algorithms to obtain *provably* robust models (Cohen et al., 2019; Gowal et al., 2019; Raghunathan et al., 2018; Salman et al., 2019; Wong & Kolter, 2018). Unfortunately, these methods either do not generalize to modern network architectures or have a prohibitively large computational complexity. As a consequence, adversarial training (Madry et al., 2018) and its variants (Alayrac et al., 2019; Carmon et al., 2019; Hendrycks et al., 2019; Kumari et al., 2019; Wu et al., 2020; Zhang et al., 2019a) have become the de facto approach to obtain robust models in practice. In essence, these methods generate adversarial examples, usually using PGD, and use them to optimize the model parameters.

While effective, adversarial training is more challenging than vanilla training. It was shown to require larger models (Xie & Yuille, 2020) and to exhibit a poor convergence behavior (Liu et al., 2020). Furthermore, as observed in (Rice et al., 2020), it suffers from *adversarial overfitting*: the robust accuracy on the test set significantly decreases in the late adversarial training phase. (Rice et al., 2020) thus proposed to perform early stopping based on a separate validation set to improve the generalization performance in adversarial training. Furthermore, (Chen et al., 2021b) introduced logit smoothing and weight smoothing strategies to reduce adversarial overfitting. In parallel to this, several techniques to improve the model's robust test accuracy were proposed (Wang et al., 2020; Wu et al., 2020; Zhang et al., 2021), but without solving the adversarial overfitting issue. By contrast, other works (Balaji et al., 2019; Huang et al., 2020) were empirically shown to mitigate adversarial overfitting but without providing any explanations as to how this phenomenon was addressed. In this paper, we study the causes of adversarial overfitting from both an empirical and a theoretical point of view. We also identify the reasons why prior attempts (Balaji et al., 2019; Chen et al., 2021a; Huang et al., 2020) successfully mitigate it.

## 3 A METRIC FOR INSTANCE DIFFICULTY

Parametric models are trained to minimize a loss objective based on several input-target pairs called training set, and are then evaluated on a held-out set called test set. By comparing the loss value of each instance, we can understand which ones, in either the training or the test set, are more difficult for the model to fit. In this section, we introduce a metric for instance difficulty, which mainly depends on the data and on the perturbations applied to the instances.

Let $\overline{\mathcal{L}}(\boldsymbol{x})$ denote the average loss of $\boldsymbol{x}$'s corresponding perturbed input across all training epochs. We define the difficulty of an instance $\boldsymbol{x}$ within a finite set $\mathcal{D}$ as

$$d(\boldsymbol{x}) = \mathbb{P}(\overline{\mathcal{L}}(\boldsymbol{x}) < \overline{\mathcal{L}}(\widetilde{\boldsymbol{x}})|\widetilde{\boldsymbol{x}} \sim U(\mathcal{D})) + \frac{1}{2}\mathbb{P}(\overline{\mathcal{L}}(\boldsymbol{x}) = \overline{\mathcal{L}}(\widetilde{\boldsymbol{x}})|\widetilde{\boldsymbol{x}} \sim U(\mathcal{D})) \,, \tag{1}$$

where $\widetilde{\boldsymbol{x}} \sim U(\mathcal{D})$ indicates $\widetilde{\boldsymbol{x}}$ is uniformly sampled from the finite set $\mathcal{D}$. $d(\boldsymbol{x})$ is a bounded function, close to 0 for the hardest instances, and 1 for the easiest ones. We discuss the motivation and properties of $d(\boldsymbol{x})$ in Appendix D.1, and show that it mainly depends on the data and the perturbation applied, the model architecture or the training duration can hardly affect the difficulty function. That is, $d(\boldsymbol{x})$ can represent the difficulty of $\boldsymbol{x}$ within a set under a specific type of perturbation.

## 4 INSTANCE DIFFICULTY AND ADVERSARIAL OVERFITTING

The model-agnostic difficulty metric of Section 3 allows us to select training instances based on their difficulty. In Figures 20 and 21 of Appendix D.2, we show some samples of the easiest and the hardest instances of each class in CIFAR10 (Krizhevsky et al., 2009) and SVHN (Netzer et al., 2011), respectively. In both cases, the easiest instances are visually highly similar, whereas the hardest ones are much more diverse, some of them being ambiguous or even incorrectly labeled. Below, we study how easy and hard instances impact the performance of adversarial training, with a focus on the adversarial overfitting phenomenon. The detailed experimental settings are deferred to Appendix C.1.

### 4.1 USING A SUBSET OF TRAINING DATA

We start by training RN18 models for 200 epochs using either the 10000 easiest, random or hardest instances of the CIFAR10 training set via either vanilla training, FGSM adversarial training or PGD

adversarial training. The adversarial budget is based on the $l_\infty$ norm and $\epsilon = 8/255$. Note that the instance's difficulty is defined under the corresponding perturbation type, and we enforce these subsets to be class-balanced. For example, the easiest 10000 instances consist of the easiest 1000 instances in each class. We provide the learning curves under different settings in Figure 1.

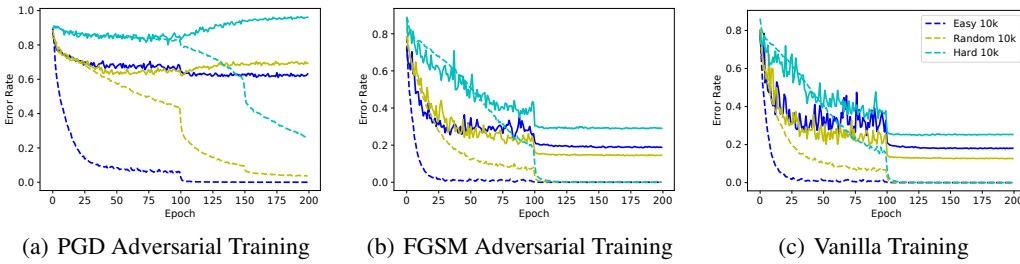

(a) PGD Adversarial Training  (b) FGSM Adversarial Training  (c) Vanilla Training

Figure 1: Learning curves obtained by training on the 10000 easiest, random and hardest instances of CIFAR10 under different scenarios. The training error (dashed lines) is the error on the selected instances, and the test error (solid lines) is the error on the whole test set.

For PGD adversarial training, in Figure 1(a), while we observe adversarial overfitting when using the random instances, as in (Rice et al., 2020), no such phenomenon occurs when using the easiest instances: the performance on the test set does not degrade during training. However, PGD adversarial training fails and suffers more severe overfitting when using the hardest instances. Note that, in Figure 5 (Appendix D.3), we show that this failure is not due to improper optimization.

In contrast to PGD, FGSM adversarial training and vanilla training (Figure 1(b), 1(c)), through which the model does not achieve true robustness (Madry et al., 2018), do not suffer from severe adversarial overfitting. In these cases, the models trained with the hardest instances also achieve non-trivial test accuracy. Furthermore, the gaps in robust test accuracy between the models trained by easy instances and by hard ones are much smaller.

In Appendix D.3, we perform additional and comprehensive experiments, evidencing that our conclusions hold for different datasets and values of $\epsilon$, and for an adversarial budget based on the $l_2$ norm. We show that more severe adversarial overfitting happens when the size of the adversarial budget $\epsilon$ increases. Furthermore, we experiment with training models using increasingly many training instances, start with the easiest ones. Our results in Figure 10 show that the models can benefit from using more data, but only using early stopping as done in (Rice et al., 2020). This indicates that the hard instances can still benefit adversarial training, but need to be utilized in an adaptive manner.

## 4.2 HARD INSTANCES LEAD TO OVERFITTING

Let us now turn to the more standard setting where we train the model with the entire training set. To nonetheless analyze the influence of instance difficulty in this scenario, we divide the training set $\mathcal{D}$ into 10 non-overlapping groups $\{\mathcal{G}_i\}_{i=0}^9$, with $\mathcal{G}_i = \{x \in \mathcal{D} | 0.1 \times i \leq d(x) < 0.1 \times (i+1)\}$. That is, $\mathcal{G}_0$ is the hardest group, whereas $\mathcal{G}_9$ is the easiest one. We then train an RN18 model on the entire CIFAR10 training set using PGD adversarial training and monitor the training behavior of the different groups. In particular, in Figure 2(a), we plot the average loss of the instances in the groups $\mathcal{G}_0$, $\mathcal{G}_3$, $\mathcal{G}_6$ and $\mathcal{G}_9$. The resulting curves show that, in the early training stages, the model first fits the easy instances, as evidenced by the average loss of group $\mathcal{G}_9$ decreasing much faster than that of the other groups. By contrast, in the late training phase, the model tries to fit the more difficult instances, with the average loss of groups $\mathcal{G}_0$ and $\mathcal{G}_3$ decreasing much faster than that of the other groups. In this period, however, the robust test error (solid grey line) increases, which indicates that adversarial overfitting arises from the model's attempt to fit the hard adversarial instances.

In addition to average losses, inspired by (Ilyas et al., 2019), which showed that the penultimate layer's activations of a robust model correspond to its *robust features* that cannot be misaligned by adversarial attacks, we monitor the group-wise average magnitudes of the penultimate layer's activations. As shown in Figure 2(b), the model first focuses on extracting robust features for the easy instances, as evidenced by the comparatively large activations of the instances in $\mathcal{G}_9$. In the late

phase of training, the norm of the activations of the hard instances increases significantly, bridging the gap between easy and hard instances. This further indicates that the model focuses more on the hard instances in the later training phase, at which point it starts overfitting.

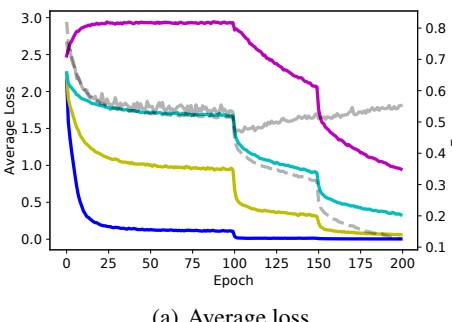

(a) Average loss.

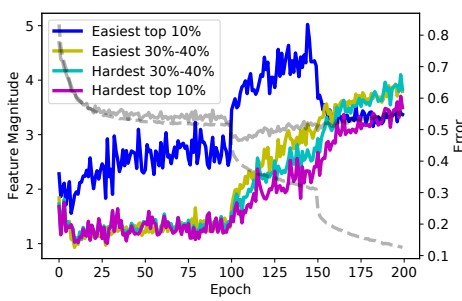

(b) Average $l_2$ norm of extracted features.

Figure 2: Analysis on the groups $\mathcal{G}_0$, $\mathcal{G}_3$, $\mathcal{G}_6$ and $\mathcal{G}_9$ in the training set. The right vertical axis corresponds to the training (dashed grey line) and test (solid grey line) error under adversarial attacks for both plots. **Left plot:** The left vertical axis represents the average loss of different groups. **Right plot:** The left vertical axis represents the average $l_2$ norm of features extracted during training for different groups.

## 5 THEORETICAL ANALYSIS

We now study the relationship between adversarial overfitting and instance difficulty from a theoretical viewpoint. We start with the linear model, where the loss function has an analytical expression, and then generalize our analysis to the nonlinear cases. We use $\{\boldsymbol{x}_i, y_i\}_{i=1}^n$ to represent the training data, and $(\mathbf{X}, \boldsymbol{y})$ as its matrix form. $\{\boldsymbol{x}'_i, y_i\}_{i=1}^n$ and $(\mathbf{X}', \boldsymbol{y})$ are their adversarial counterparts. Here, $\boldsymbol{x}_i \in \mathbb{R}^m$, $y_i \in \{-1, +1\}$, $\mathbf{X} \in \mathbb{R}^{n \times m}$ and $\boldsymbol{y} \in \{-1, +1\}^n$. The notation used for our theoretical analysis is summarized in Table 3 of Appendix A.

### 5.1 LOGISTIC REGRESSION

We study the logistic regression model under an $l_2$ norm based adversarial budget. In this case, the model is parameterized by $\boldsymbol{w} \in \mathbb{R}^m$ and outputs $sign(\boldsymbol{w}^T \boldsymbol{x}'_i)$ given the adversarial example $\boldsymbol{x}'_i$ of the input $\boldsymbol{x}_i$. The loss function for this instance is $\frac{1}{1+e^{y_i \boldsymbol{w}^T \boldsymbol{x}'_i}}$. We assume over-parameterization, which means $n < m$.

The following theorem shows that, under mild assumptions, the parameters of the adversarially trained logistic regression model converge to the $l_2$ max-margin direction of the training data.

**Theorem 1.** *For a dataset $\{\boldsymbol{x}_i, y_i\}_{i=1}^n$ that is linearly separable under the adversarial budget $\mathcal{S}^{(2)}(\epsilon)$, any initial point $\boldsymbol{w}_0$ and step size $\alpha \leq 2\|\mathbf{X}\|^{-2}$, the gradient descent $\boldsymbol{w}_{u+1} = \boldsymbol{w}_u - \alpha \nabla_{\boldsymbol{w}} \mathcal{L}_{\boldsymbol{w}_u}(\mathbf{X}')$ converges asymptotically to the $l_2$ max-margin vector of the training data. That is,*

$$\lim_{u \to \infty} \frac{\boldsymbol{w}_u}{\|\boldsymbol{w}_u\|} = \frac{\widehat{\boldsymbol{w}}}{\|\widehat{\boldsymbol{w}}\|}, \text{ where } \widehat{\boldsymbol{w}} = \arg\min_{\boldsymbol{w}} \|\boldsymbol{w}\| \quad s.t. \ \forall i \in \{1, 2, ..., n\}, \ \boldsymbol{w}^T \boldsymbol{x}_i \geq 1. \quad (2)$$

The proof is deferred to Appendix B.1. Theorem 1 extends the conclusion in (Soudry et al., 2018), which only studies the non-adversarial case. It also indicates that the optimal parameters are only determined by the support vectors of the training data, which are the ones with the smallest margin. According to the loss function, the smallest margin means the largest loss values and thus the hardest training instances based on our definition in Section 3.

To further study how the training instances' difficulty influences the model's generalization performance, we assume that the data points are drawn from a $K$-mode Gaussian mixture model (GMM). Specifically, the $k$-th component has a probability $p_k$ of being sampled and is formulated as follows:

$$\text{if } y_i = +1, \ \boldsymbol{x}_i \sim \mathcal{N}(r_k \boldsymbol{\eta}, \mathbf{I}); \text{ if } y_i = -1, \ \boldsymbol{x}_i \sim \mathcal{N}(-r_k \boldsymbol{\eta}, \mathbf{I}). \quad (3)$$

Here, $\boldsymbol{\eta} \in \mathbb{R}^m$ is the unit vector indicating the mean of the positive instances, and $r_k \in \mathbb{R}^+$ controls the average distance between the positive and negative instances. The mean values of all modes in this GMM are colinear, so $r_k$ indicates the difficulty of instances sampled from the $k$-th component. Without the loss of generality, we assume $r_1 < r_2 < ... < r_{K-1} < r_K$. As in Section 4, we consider models trained with the subsets of the training data, e.g., $n$ instances from the $l$-th component. $l = 1$ then indicates training on the hardest examples, while $l = K$ means using the easiest. In matrix form, we have $\mathbf{X} = r_l \boldsymbol{y}\boldsymbol{\eta}^T + \mathbf{Q}$ for the instances sampled from the $l$-th component, where the rows of noise matrix $\mathbf{Q}$ are sampled from $\mathcal{N}(\mathbf{0}, \mathbf{I})$.

Although the max-margin direction in Equation (2), where the parameters converge based on Theorem 1, does not have an analytical expression, the results in (Wang & Thrampoulidis, 2020) indicate that, in the over-parameterization regime and when the training data is sampled from a GMM, the max-margin direction is the min-norm interpolation of the data with high probability. Since the latter has an analytical form given by $\mathbf{X}^T(\mathbf{X}\mathbf{X}^T)^{-1}\boldsymbol{y}$, we can then calculate the exact generalization performance of the trained model as stated in the following theorem.

**Theorem 2.** *If a logistic regression model is adversarially trained on $n$ separable training instances sampled from the $l$-th component of the GMM model described in (3). If $\frac{m}{n \log n}$ is sufficiently large[2], then with probability $1 - O(\frac{1}{n})$, the expected adversarial test error $\mathcal{R}$ under the adversarial budget $\mathcal{S}^{(2)}(\epsilon)$, which is a function of $r_l$ and $\epsilon$, on the whole GMM model described in (3) is given by*

$$\mathcal{R}(r_l, \epsilon) = \sum_{k=1}^{K} p_k \Phi\left(r_k g(r_l) - \epsilon\right), \ g(r_l) = \left(C_1 - \frac{1}{C_2 r_l^2 + o(r_l^2)}\right)^{\frac{1}{2}}, \ C_1, C_2 \geq 0. \quad (4)$$

*$C_1, C_2$ are independent of $\epsilon$ and $r_l$. The function $\Phi$ is defined as $\Phi(x) = \mathbb{P}(Z > x), \ Z \sim \mathcal{N}(0, 1)$.*

We defer the proof of Theorem 2 to Appendix B.2, in which we calculate the *exact* expression of $\mathcal{R}(r_l, \epsilon)$, $C_1$, $C_2$, and show that $C_1$, $C_2$ are positive numbers almost surely. Since $C_1$ and $C_2$ are independent of $r_l$, and $\Phi(x)$ is a monotonically decreasing function, we conclude that the robust test error $\mathcal{R}(r_l, \epsilon)$ becomes smaller when $r_l$ increases. That is, when the training instances become easier, the corresponding generalization error under the adversarial attack becomes smaller.

Theorem 2 holds for all $\epsilon$ only if the training data is separable under the corresponding adversarial budget. The following corollary shows that the difference in the robust test error between models trained with easy instances and the ones with hard ones increases when $\epsilon$ becomes larger.

**Corollary 1.** *Under the conditions of Theorem 2 and the definition of $\mathcal{R}$ in Equation (4), if $\epsilon_1 < \epsilon_2$, then we have $\forall \, 0 \leq i < j \leq K, \mathcal{R}(r_i, \epsilon_1) - \mathcal{R}(r_j, \epsilon_1) < \mathcal{R}(r_i, \epsilon_2) - \mathcal{R}(r_j, \epsilon_2)$.*

The proof is in Appendix B.3. This indicates that, compared with training on clean inputs, i.e., $\epsilon = 0$, the generalization performance of adversarial training with $\epsilon > 0$ is more sensitive to the difficulty of the training instances. This is consistent with our empirical observations in Figure 1.

## 5.2 General Nonlinear Models

In this section, we study the binary classification problem using a general nonlinear model. We consider a model with $b$ parameters, i.e., $\boldsymbol{w} \in \mathbb{R}^b$. Without loss of generality, we assume the output of the function $f_{\boldsymbol{w}}$ to lie in $[-1, +1]$. Furthermore, we assume isoperimetry of the data distribution:

**Assumption 1.** *The data distribution $\mu$ is a composition of $K$ $c$-isoperimetric distributions on $\mathbb{R}^m$, each of which has a positive conditional variance. That is, $\mu = \sum_{k=1}^{K} \alpha_k \mu_k$, where $\alpha_k > 0$ and $\sum_{k=1}^{K} \alpha_k = 1$. We define $\sigma_k^2 = \mathbb{E}_{\mu_k}[Var[y|\boldsymbol{x}]]$, and without loss of generality assume that $\sigma_1 \geq \sigma_2 \geq ... \geq \sigma_K > 0$. Furthermore, given any $L$-Lipschitz function $f_{\boldsymbol{w}}$, i.e., $\forall \boldsymbol{x}_1, \boldsymbol{x}_2, \|f_{\boldsymbol{w}}(\boldsymbol{x}_1) - f_{\boldsymbol{w}}(\boldsymbol{x}_2)\| \leq L\|\boldsymbol{x}_1 - \boldsymbol{x}_2\|$, we have*

$$\forall k \in \{1, 2, ..., K\} \ \mathbb{P}(\boldsymbol{x} \sim \mu_k, \|f_{\boldsymbol{w}}(\boldsymbol{x}) - \mathbb{E}_{\mu_k}(f_{\boldsymbol{w}})\| \geq t) \leq 2e^{-\frac{mt^2}{2cL^2}} \ . \quad (5)$$

This is a benign assumption; the data distribution is a mixture of $K$ components and each of them contains samples from a sub-Gaussian distribution. These components correspond to training

---

[2] Specifically, $m$ and $n$ need to satisfy $m > 10n \log n + n - 1$ and $m > Cnr_l\sqrt{\log 2n}\|\boldsymbol{\eta}\|$. The constant $C$ is derived in the proof of Theorem 1 in (Wang & Thrampoulidis, 2020).

instances of different difficulty levels measured by the conditional variance. We then study the property of the model $f_{\boldsymbol{w}}$ under adversarial attacks.

**Definition 1.** *Given the dataset $\{\boldsymbol{x}_i, y_i\}_{i=1}^n$, the model $f_{\boldsymbol{w}}$, the adversarial budget $\mathcal{S}^{(p)}(\epsilon)$ and a positive constant $C$, we define the function $h(C, \epsilon)$ as*

$$h(C, \epsilon) = \min_{\boldsymbol{w} \in \mathcal{T}(C, \epsilon)} \min_i h_{i, \boldsymbol{w}}(\epsilon) \ \ s.t. \ \mathcal{T}(C, \epsilon) = \left\{ \boldsymbol{w} | \frac{1}{n} \sum_{i=1}^n (f_{\boldsymbol{w}}(\boldsymbol{x}_i') - y_i)^2 \leq C \right\} , \quad (6)$$

*where $h_{i, \boldsymbol{w}}(\epsilon) = \max \zeta, \ s.t. \ [f_{\boldsymbol{w}}(\boldsymbol{x}_i) - \zeta, f_{\boldsymbol{w}}(\boldsymbol{x}_i) + \zeta] \subset \left\{ f_{\boldsymbol{w}}(\boldsymbol{x}_i + \Delta) | \Delta \in \mathcal{S}^{(p)}(\epsilon) \right\}.$*

*Here, $\boldsymbol{x}_i'$ is the adversarial example of $\boldsymbol{x}$. We omit the superscript $(p)$ for notation simplicity.*

By definition, $h_{i, \boldsymbol{w}}(\epsilon) \geq 0$ depicts the bandwidth $\zeta$ of the model's output range in the domain of the adversarial budget on a training instance. $h(C, \epsilon)$ is the minimum bandwidth among the models whose mean squared error on the adversarial training set is smaller than $C$. Based on the definitions of $\mathcal{T}$ and $h_{i, \boldsymbol{w}}$, and for a fixed value of $C$, we have $\forall \epsilon_1 < \epsilon_2, h_{i, \boldsymbol{w}}(\epsilon_1) \leq h_{i, \boldsymbol{w}}(\epsilon_2)$ and $\mathcal{T}(C, \epsilon_2) \subset \mathcal{T}(C, \epsilon_1)$. As a result, $\forall \epsilon_1 < \epsilon_2, h(C, \epsilon_1) \leq h(C, \epsilon_2)$. In addition, since $\forall C_1 < C_2$, $\mathcal{T}(C_1, \epsilon) \subset \mathcal{T}(C_2, \epsilon)$ for a fixed value of $\epsilon$, we have $\forall C_1 < C_2, h(C_1, \epsilon) \geq h(C_2, \epsilon)$. That is to say, $h(C, \epsilon)$ is a monotonically non-decreasing function on $\epsilon$ and a monotonically non-increasing function on $C$. In practice, when $f_{\boldsymbol{w}}$ represents a deep neural network, $h(C, \epsilon)$ increases with $\epsilon$ almost surely, because the attack algorithm usually generates adversarial examples at the boundary of the adversarial budget. We then state our main theorem below.

**Theorem 3.** *Given $n$ training pairs $\{\boldsymbol{x}_i, y_i\}_{i=1}^n$ sampled from the $l$-th component $\mu_l$ of the distribution in Assumption 1, the parametric model $f_{\boldsymbol{w}}$, the adversarial budget $\mathcal{S}^{(p)}(\epsilon)$ and the corresponding function $h$ defined in Definition 1, we assume that the model $f_{\boldsymbol{w}}$ is in the function space $\mathcal{F} = \{f_{\boldsymbol{w}}, \boldsymbol{w} \in \mathcal{W}\}$ with $\mathcal{W} \subset \mathbb{R}^b$ having a finite diameter $diam(\mathcal{W}) \leq W$ and, $\forall \boldsymbol{w}_1, \boldsymbol{w}_2 \in \mathcal{W}, \|f_{\boldsymbol{w}_1} - f_{\boldsymbol{w}_2}\|_\infty \leq J\|\boldsymbol{w}_1 - \boldsymbol{w}_2\|_\infty$. We train the model $f_{\boldsymbol{w}}$ adversarially using these $n$ data points. Let $\boldsymbol{x}'$ be the adversarial example of the data point $\boldsymbol{x}$ and $\delta \in (0, 1)$. If we have $\frac{1}{n} \sum_{i=1}^n (f_{\boldsymbol{w}}(\boldsymbol{x}_i') - y_i)^2 = C$ and $\gamma := \sigma_l^2 + h^2(C, \epsilon) - C \geq 0$, then with probability at least $1 - \delta$, the Lipschitz constant of $f_{\boldsymbol{w}}$ is lower bounded as*

$$Lip(f_{\boldsymbol{w}}) \geq \frac{\gamma}{2^7} \sqrt{\frac{nm}{c \left( b \log(4WJ\gamma^{-1}) - \log(\delta/2 - 2e^{-2^{-11} n \gamma^2}) \right)}} , \quad (7)$$

*where $Lip(f_{\boldsymbol{w}})$ is the Lipschitz constant of $f_{\boldsymbol{w}}$: $\forall \boldsymbol{x}_1, \boldsymbol{x}_2, \|f_{\boldsymbol{w}}(\boldsymbol{x}_1) - f_{\boldsymbol{w}}(\boldsymbol{x}_2)\| \leq Lip(f_{\boldsymbol{w}})\|\boldsymbol{x}_1 - \boldsymbol{x}_2\|.$*

The proof is deferred to Appendix B.4. Theorem 3 extends the results in (Bubeck & Sellke, 2021) to the case of adversarial training. Note that modern deep neural network models typically have millions of parameters, so $b \gg \max\{c, m, n\}$. In this case, we can approximate the lower bound (7) by $Lip(f_{\boldsymbol{w}}) \gtrsim \frac{\gamma}{2^7} \sqrt{\frac{nm}{bc \log(4WJ\gamma^{-1})}}$, and the right hand side increases with $\gamma$. Since $\gamma := \sigma_l^2 + h^2(C, \epsilon) - C$, the lower bound increases with both $\sigma_l$ and $\epsilon$ but decreases as $C$ increases.

The Lipschitz constant is widely used to bound a model's adversarial vulnerability (Ruan et al., 2018; Weng et al., 2018a;b): larger Lipschitz constants indicate higher adversarial vulnerability. Recall that $\gamma$ needs to be non-negative, so $C$ is upper bounded, which means that the model is well fit to the adversarial training set and the adversarial vulnerability is approximately the generalization gap. Based on this, the adversarial vulnerability of a model increases with the size of the adversarial budget and the difficulty level of the training instances; it also increases as the training mean squared error decreases. That is, under the same adversarial budget, the adversarial vulnerability increases with the instances' difficulty, measured by $\sigma_l$ in our theorem; using the same training instances, the adversarial vulnerability increases with the adversarial budget measured by $\epsilon$. In addition, as adversarial training progresses, the mean squared error $C$ on the adversarial training instances becomes smaller, which makes the lower bound of the Lipschitz constant larger. This indicates that adversarial vulnerability becomes larger in the later phase of adversarial training.

We provide empirical evidence to confirm the conclusions of Theorem 3 in Appendix D.4. Since calculating the Lipschitz constant of a deep neural network is NP-hard (Scaman & Virmaux, 2018), exactly calculating the Lipschitz constant (Jordan & Dimakis, 2020) can only be achieved for simple multi-layer perceptron (MLP) models, not for modern deep networks. Instead, we estimate the upper bound of the Lipschitz constant numerically, as in (Scaman & Virmaux, 2018).

## 6 Mitigating Adversarial Overfitting

Existing methods mitigating adversarial overfitting study the standard adversarial training scenario: PGD adversarial training from scratch. In Appendix D.5, we show that they use either adaptive inputs (Balaji et al., 2019) or adaptive targets (Huang et al., 2020; Chen et al., 2021b) to implicitly avoid fitting hard input-target pairs, thus providing an explanation for their success.

Below, we study two settings other than standard adversarial training: fast adversarial training and adversarial fine-tuning with additional data, to validate our findings. We show that methods inspired by our findings can avoid adversarial overfitting and improve the performance. The detailed experimental settings for this section are in Appendix C.2.

### 6.1 Fast Adversarial Training

Adversarial training in (Madry et al., 2018) introduces a significant computational overhead. Thus it is desirable to accelerate this method. Our experiments in this section are based on adversarial training with transferable adversarial examples (ATTA in (Zheng et al., 2020)), which stores the adversarial perturbation for each training instance as an initial point for the next epoch. We show that adaptively utilizing the easy and hard training instances not only mitigates adversarial overfitting, but also significantly improves the performance of the final model.

First, we introduce a reweighting scheme to assign lower weights to hard instances when calculating the loss objective. Specifically, each training instance is assigned a weight equal to the adversarial output probability of the true label. Then this weight is normalized to ensure that the weights in a mini-batch sum to $1$. Note that our reweighting scheme is based on the adversarial output instead of the clean output, because the adversarial output probability will also be used to calculate the loss objective. As a result, the computational overhead of the reweighting scheme is negligible.

In addition to reweighting, we also use adaptive targets to improve the performance. For each training instance $(\boldsymbol{x}, y)$, we maintain an adaptive moving average target $\widetilde{\boldsymbol{t}}$. $\widetilde{\boldsymbol{t}}$ is updated in an exponential average manner in each epoch $\widetilde{\boldsymbol{t}} \leftarrow \rho \widetilde{\boldsymbol{t}} + (1 - \rho)\boldsymbol{o}'$ where $\rho$ is the momentum factor. This is similar to the target in (Huang et al., 2020), but, similarly to the reweighting scheme, we use the adversarial output $\boldsymbol{o}'$ instead of the clean output $\boldsymbol{o}$ to avoid an increase in computational complexity. The final adaptive target we use is $\boldsymbol{t} = \beta \mathbf{1}_y + (1 - \beta)\widetilde{\boldsymbol{t}}$ and thus the loss objective is $\mathcal{L}_{\boldsymbol{w}}(\boldsymbol{x}', \boldsymbol{t})$. The factor $\beta$ controls how "adaptive" our target is: $\beta = 0$ yields a fully adaptive moving average target $\widetilde{\boldsymbol{t}}$ and $\beta = 1$ yields a one-hot target $\mathbf{1}_y$. We provide the pseudocode as Algorithm 1 in Appendix C.2.

We run experiments on CIFAR10 using WRN34 models under the $l_\infty$ adversarial budget of size $\epsilon = 8/255$, the standard setting where most fast adversarial training algorithms are benchmarked (Croce et al., 2020). We evaluate the model's robust accuracy on the test set by AutoAttack (Croce & Hein, 2020b), the popular and reliable attack for evaluation. The results are provided in Table 2, where the results of the baseline methods are taken from RobustBench (Croce et al., 2020). We also report the number of epochs and the number of forward and backward passes in a mini-batch update of each method. The product of these two values indicates the training complexity.

We can clearly see that both reweighting and adaptive targets improve the performance on top of ATTA (Zheng et al., 2020). Note that our method based on adaptive targets achieve the best performance while needing only $1/4$ of the training time of (Chen et al., 2021a), the strongest baseline. (Wong et al., 2020) is the only baseline consuming less training time than ours, but its performance is much worse than ours; it suffers from catastrophic overfitting when using a WRN34 model. In Appendix D.6, we provide the learning curves of our methods under different settings and show that both reweighting and adaptive targets mitigate adversarial overfitting. We also conduct an ablation study on the value of $\beta$ and find that a decrease in $\beta$ decreases the generalization gap. This indicates that the more adaptive the targets, the smaller the generalization gap.

### 6.2 Adversarial Finetuning with Additional Data

We observe that adversarial overfitting occurs in the small learning rate regime in Section 4. To further study this, we propose to fine-tune an adversarially pretrained model using additional training data, because we also use small learning rate to fine-tune a model. While additional training data was

| Method | Model | Epochs | Complexity | AA |
|--------|-------|--------|------------|-----|
| (Shafahi et al., 2019) | WRN34 | 200 | 2 | 41.17 |
| (Wong et al., 2020) | RN18 | 15 | 4 | 43.21 |
| (Zheng et al., 2020) | WRN34 | 38 | 4 | 44.48 |
| (Zhang et al., 2019a) | WRN34 | 105 | 3 | 44.83 |
| (Chen et al., 2021a) | WRN34 | 100 | 7 | 51.12 |
| Reweighting (Ours) | WRN34 | 38 | 4 | 46.15 |
| Adaptive Target (Ours) | WRN34 | 38 | 4 | 51.17 |

Table 1: Comparison between different accelerated adversarial training methods in robust test accuracy against AutoAttack (AA). The baseline results are from RobustBench. *Complexity* shows the number of forward passes and backward passes in one mini-batch update.

| Duration | Method | AA |
|----------|--------|-----|
| **WRN34 on CIFAR10, $\epsilon = 8/255$** | | |
| No Fine Tuning | | 52.01 |
| 1 Epoch | Vanilla AT | 54.11 |
| | Ours | 54.69 |
| 5 Epoch | Vanilla AT | 55.49 |
| | Ours | 56.99 |
| **RN18 on SVHN, $\epsilon = 0.02$** | | |
| No Fine Tuning | | 67.77 |
| 1 Epoch | Vanilla AT | 70.81 |
| | Ours | 72.53 |
| 5 Epoch | Vanilla AT | 72.18 |
| | Ours | 73.35 |

Table 2: Robust accuracy of fine-tuned models against AutoAttack(AA).

shown to be beneficial in (Alayrac et al., 2019; Carmon et al., 2019), we demonstrate that letting the model adaptively fit the easy and hard instances can further improve the performance.

We conduct experiments on both CIFAR10 and SVHN, using WRN34 and RN18 models, respectively. The experimental settings are the same as (Carmon et al., 2019) except the learning rate. We tune the learning rate and find that fixing it to $10^{-3}$ is the best choice. The model is fine-tuned for either $1$ epoch or $5$ epochs, which means that each additional training instance is used either $5$ times or only once. This is because we observed the performance of vanilla adversarial training to start decaying after $5$ epochs. As such, methods requiring many epochs such as (Balaji et al., 2019) and (Huang et al., 2020) are not applicable here.

Our first technique, reweighting, is the same as in Section 6.1. In addition to reweighting, we can also add a KL regularization term measuring the KL divergence between the output probability of the clean instance and of the adversarial instance. The KL term encourages the adversarial output to be close to the clean one. In other words, the clean output probability serves as the adaptive target. For hard instances, the clean and adversarial inputs are usually both misclassified. Therefore, the clean outputs of these instances constitute simpler targets compared with the ground-truth labels. Ultimately, the loss objective of a mini-batch $\{x_i\}_{i=1}^B$ used for fine-tuning is expressed as $\mathcal{L}_{FT}(\{x_i\}_{i=1}^B) = \sum_{i=1}^B w_i \left[\mathcal{L}_w(x_i') + \lambda KL(o_i||o_i')\right]$ where $w_i$ is the adaptive weight when we use re-weighting, or $1/B$ otherwise. $\lambda$ is 6 when using the regularization term and 0 otherwise.

We use both reweighting and KL regularization to fine-tune the model. Our results are shown in Table 2, where the robust test accuracy is also evaluated by AutoAttack. It is clear that our methods can improve the performance under all settings. We also conduct an ablation study in Appendix D.7. All these results show that avoiding fitting hard adversarial examples helps to improve the generalization performance in adversarial fine-tuning with additional training data.

# 7 CONCLUSION

We have investigated *adversarial overfitting* from the perspective of the easy and hard training instances. By introducing a quantitative, model-agnostic and normalized metric to measure the instance difficulty, we have shown that a model's generalization performance under adversarial attacks degrades during the later phase of training as the model fits the hard adversarial instances. We have conducted theoretical analyses on both linear and nonlinear models. On an over-parameterized logistic regression model, we have shown that training on harder adversarial instances leads to poorer generalization performance. On general nonlinear models, we have proven that the lower bound of a well-trained model's Lipschitz constant increases with the difficulty of the training instances. Finally, our experiments on fast adversarial training and adversarial fine-tuning with additional data have demonstrated that adaptively using the training instances mitigates adversarial overfitting and improves the model's robustness.

## ETHICS STATEMENT

Adversarial attacks may cause unexpected failure of modern deep learning models, especially in some safety-critical applications such as self-driving and medical imaging. Our work studies the defense methods and focuses on how to improve training robust models to prevent such failure from happening. We are not aware of any ethics issues about our work, as we are using publicly available data and models.

## REPRODUCIBILITY STATEMENT

The experimental settings of our empirical study are demonstrated in detail in Appendix C. Especially, we also provide the pseudocode of our algorithm there. Our code is submitted anonymously on GoogleDrive[3]. It will be made publicly available when our work is published.

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

## A  NOTATION

| | | |
|---|---|---|
| $b$ | Section 5.2 | The number of parameters in a general nonlinear model. |
| $c$ | Assumption 1, Section 5.2 | The coefficient in isoperimetry. |
| $C$ | Section 5.2 | The mean squared error on the adversarial training set. |
| $d$ | Equation 1, Section 3 | The function introduced by the proposed difficulty metric. |
| $\mathcal{D}$ | Section 3 | The data set. |
| $f_{\boldsymbol{w}}$ | Section 1 | The model parameterized by $\boldsymbol{w}$. |
| $\mathcal{F}$ | Theorem 3, Section 5.2 | The function space of the model. |
| $\mathcal{G}$ | Section 4.2 | Groups of the training set divided by instance difficulty. |
| $h$ | Definition 1, Section 5.2 | The bandwidth of the model's output range. |
| $J$ | Theorem 3, Section 5.2 | The Lipschitz constant of $f_{\boldsymbol{w}}$ w.r.t $\boldsymbol{w}$. |
| $K$ | Section 5 | The number of components in the data distribution. |
| $l$ | Section 5 | The component index where the training data is sampled. |
| $L$ | Assumption 1, Section 5.2 | The Lipschitz constant of $f_{\boldsymbol{w}}$ w.r.t the input. |
| $\mathcal{L}$ | Section 1 | The loss function. |
| $m$ | Section 5 | Dimension of the input data. |
| $n$ | Section 5 | The number of training instances. |
| $\boldsymbol{o}, \boldsymbol{o}'$ | Section 6 | Model's output of the clean and the adversarial input. |
| $p$ | Section 1 | Shape of the adversarial budget. |
| $r$ | Equation 3, Section 5.1 | The coefficient in the GMM model. |
| $\mathcal{R}$ | Theorem 2, Section 5.1 | The robust test error. |
| $\boldsymbol{t}, \widetilde{\boldsymbol{t}}$ | Section 6.1 | The adaptive target and the moving average target. |
| $\boldsymbol{w}$ | Section 5 | Model parameters. |
| $W$ | Theorem 3, Section 5.2 | The diameter upper bound of the parameter space. |
| $\mathcal{W}$ | Theorem 3, Section 5.2 | The space of model parameters. |
| $\boldsymbol{x}, \boldsymbol{x}', \mathbf{X}$ | Section 1 & Section 5 | Clean input, adversarial input and its matrix form. |
| $y, \boldsymbol{y}$ | Section 1 & Section 5 | Label and its vector form. |
| $\alpha$ | Algorithm 1 | The step size of the adversarial attacks. |
| $\beta$ | Section 6.1 | The coefficient controlling how adaptive the target is. |
| $\gamma$ | Theorem 3, Section 5.2 | The non-negative variable introduced in Theorem 3. |
| $\delta$ | Theorem 3, Section 5.2 | The probability introduced in Theorem 3. |
| $\epsilon$ | Section 1 | The size of the adversarial budget. |
| $\boldsymbol{\eta}$ | Equation 3, Section 5.1 | The direction of the mean of each GMM's component. |
| $\rho$ | Section 6.1 | The momentum calculating the moving average target. |
| $\mu_l, \mu_l$ | Assumption 1, Section 5.2 | Data distribution and its $l$-th component. |
| $\sigma$ | Assumption 1, Section 5.2 | The conditional variance of the data distribution. |

Table 3: The notation in this paper. We provide the section of their definition or first appearance.

## B  PROOFS IN THEORETICAL ANALYSIS

### B.1  PROOF OF THEOREM 1

Similar to (Soudry et al., 2018), we can assume all instances are positive without the loss of generality, this is because we can always redefine $y_i \boldsymbol{x}_i$ as the input. In this regard, the loss to optimize in a logistic regression model under the adversarial budget $\mathcal{S}^{(2)}(\epsilon)$ is:

$$\mathcal{L}_{\boldsymbol{w}}(\mathbf{X}) = \sum_{i=1}^{n} l(\boldsymbol{w}^T \boldsymbol{x}_i - \epsilon \|\boldsymbol{w}\|) \tag{8}$$

Here $l(\cdot)$ is the logistic function: $l(x) = \frac{1}{1+e^{-x}}$. We use $\mathbf{X} \in \mathbb{R}^{n \times m}$ to represent the training set as said in Section 5, then the loss function $\mathcal{L}(\boldsymbol{w})$ is $\|\mathbf{X}\|^2$-smooth, where $\|\mathbf{X}\|^2$ is the maximal singular value of $\mathbf{X}$. Since function $\mathcal{L}_{\boldsymbol{w}}$ is convex on $\boldsymbol{w}$, so gradient descent of step size smaller than $2\|\mathbf{X}\|^{-2}$ will asymptotically converge to the global infimum of the function $\mathcal{L}_{\boldsymbol{w}}$ on $\boldsymbol{w}$.

Before proving Theorem 1, we first introduce the following lemma:

**Lemma 1.** *Consider the max-margin vector $\widehat{\boldsymbol{w}}$ of the vanilla case defined in Equation (2), we then introduce the max margin vector $\widehat{\boldsymbol{w}'}$ defined under the adversarial attack of budget $\mathcal{S}^{(2)}(\epsilon)$ as follows:*

$$\widehat{\boldsymbol{w}'} = \arg\min_{\boldsymbol{w}} \|\boldsymbol{w}\| \ \ s.t. \ \forall i \in \{1, 2, ..., n\}, \ \boldsymbol{w}^T \boldsymbol{x}_i - \epsilon \|\boldsymbol{w}\| \geq 1 \tag{9}$$

*Then we have $\widehat{\boldsymbol{w}'}$ is collinear with $\widehat{\boldsymbol{w}}$, i.e., $\frac{\widehat{\boldsymbol{w}'}}{\|\widehat{\boldsymbol{w}'}\|} = \frac{\widehat{\boldsymbol{w}}}{\|\widehat{\boldsymbol{w}}\|}$*

*Proof.* We show that $\widehat{\boldsymbol{w}} = \frac{1}{1+\epsilon\|\widehat{\boldsymbol{w}'}\|}\widehat{\boldsymbol{w}'}$ and prove it by contraction.

Let's assume $\exists \boldsymbol{v}$, $s.t.$ $\|\boldsymbol{v}\| < \frac{\|\widehat{\boldsymbol{w}'}\|}{1+\epsilon\|\widehat{\boldsymbol{w}'}\|}$ and $\forall i \in \{1, 2, ..., n\}$, $\boldsymbol{v}^T \boldsymbol{x}_i \geq 1$, then we can consider $\boldsymbol{v}' = (1 + \|\widehat{\boldsymbol{w}'}\|)\boldsymbol{v}$. The $l_2$ norm of $\boldsymbol{v}'$ is smaller than that of $\widehat{\boldsymbol{w}'}$, and we have

$$\forall i \in \{1, 2, ..., n\}, \boldsymbol{v}'^T \boldsymbol{x}_i - \epsilon \|\boldsymbol{v}'\| = (1 + \epsilon\|\widehat{\boldsymbol{w}'}\|)\boldsymbol{v}^T \boldsymbol{x}_i - \epsilon\|\boldsymbol{v}'\| > (1 + \epsilon\|\widehat{\boldsymbol{w}'}\|) - \epsilon\|\widehat{\boldsymbol{w}'}\| = 1 \tag{10}$$

Inequality 10 shows we can construct a vector $\boldsymbol{v}'$ whose $l_2$ norm is smaller than $\widehat{\boldsymbol{w}'}$ and satisfying the condition (9), this contracts with the optimality of $\widehat{\boldsymbol{w}'}$. Therefore, there is no solution of condition (2) whose norm is smaller than $\frac{\|\widehat{\boldsymbol{w}'}\|}{1+\epsilon\|\widehat{\boldsymbol{w}'}\|}$.

On the other hand, $\frac{1}{1+\epsilon\|\widehat{\boldsymbol{w}'}\|}\widehat{\boldsymbol{w}'}$ satisfies the condition (2) and its $l_2$ norm is $\frac{\|\widehat{\boldsymbol{w}'}\|}{1+\epsilon\|\widehat{\boldsymbol{w}'}\|}$. As a result, we have $\widehat{\boldsymbol{w}} = \frac{1}{1+\epsilon\|\widehat{\boldsymbol{w}'}\|}\widehat{\boldsymbol{w}'}$. That means $\widehat{\boldsymbol{w}}$ and $\widehat{\boldsymbol{w}'}$ are collinear. $\square$

With Lemma 1, Theorem 1 is more straightforward, whose proof is shown below. Regarding the convergence analysis of the logistic regression model in non-adversarial cases, we encourage the readers to find more details in (Ji & Telgarsky, 2019; Soudry et al., 2018).

*Proof.* Theorem 1 in (Ji & Telgarsky, 2019) and Theorem 3 in (Soudry et al., 2018) proves the convergence of the direction of the logistic regression parameters in different cases. In this regard, we can let $\boldsymbol{w}_\infty = \lim_{u\to\infty} \frac{\boldsymbol{w}(u)}{\|\boldsymbol{w}(u)\|}$. That is to say, for sufficiently large $u$, the direction of the parameter $\boldsymbol{w}(u)$ can be considered fixed. As a result, the adversarial perturbations of each data instance $\boldsymbol{x}_i$ is fixed, i.e., $\epsilon\boldsymbol{w}_\infty$.

We can then apply the conclusion of Theorem 3 in (Soudry et al., 2018) here, the only difference is the data points are $\{\boldsymbol{x}_i - \epsilon\boldsymbol{w}_\infty\}_{i=1}^n$. Therefore, the parameter $\boldsymbol{w}(u)$ will converge to the $l_2$ max margin of the dataset $\{\boldsymbol{x}_i - \epsilon\boldsymbol{w}_\infty\}_{i=1}^n$. When $t \to \infty$, we have $\boldsymbol{w}(u)^T(\boldsymbol{x}_i - \epsilon\boldsymbol{w}_\infty) = \boldsymbol{w}(u)^T\boldsymbol{x}_i - \epsilon\|\boldsymbol{w}(u)\|$. This is exactly the adversarial max margin condition in (9). Based on Lemma 1, we have $\lim_{u\to\infty} \frac{\boldsymbol{w}(u)}{\|\boldsymbol{w}(u)\|} = \frac{\widehat{\boldsymbol{w}'}}{\|\widehat{\boldsymbol{w}'}\|} = \frac{\widehat{\boldsymbol{w}}}{\|\widehat{\boldsymbol{w}}\|}$ $\square$

## B.2 Proof of Theorem 2

Given the parameter $\boldsymbol{w}$ of the logistic regression model, we can first calculate the robust error for the $k$-th component of the GMM model defined in (3).

**Lemma 2.** *The 0-1 classification error of a linear classifier $\boldsymbol{w}$ under the adversarial attack of the budget $\mathcal{S}^{(2)}(\epsilon)$ for the $k$-th component of the GMM model defined in (3) is:*

$$\widehat{\mathcal{R}}_k(\epsilon) = \Phi\left(\frac{r_k \boldsymbol{w}^T \boldsymbol{\eta}}{\|\boldsymbol{w}\|} - \epsilon\right) \tag{11}$$

*where $\Phi(x) = \mathbb{P}(Z > x), Z \sim \mathcal{N}(0, 1)$.*

*Proof.* For a random drawn data instance $(\boldsymbol{x}, y)$, the adversarial perturbation is $-y\epsilon\frac{\boldsymbol{w}}{\|\boldsymbol{w}\|}$. Let's decompose $\boldsymbol{x}$ as $r_k y\boldsymbol{\eta} + \boldsymbol{z}$, where $\boldsymbol{z} \sim \mathcal{N}(0, \mathbf{I})$. Then, we have

$$\widehat{\mathcal{R}}_k(\epsilon) = \mathbb{P}(y\boldsymbol{w}^T(\boldsymbol{x} - y\epsilon\frac{\boldsymbol{w}}{\|\boldsymbol{w}\|}) < 0) = \mathbb{P}(y\boldsymbol{w}^T(r_k y\boldsymbol{\eta} + \boldsymbol{z} - y\epsilon\frac{\boldsymbol{w}}{\|\boldsymbol{w}\|}) < 0)$$
$$= \mathbb{P}(-y\boldsymbol{w}^T\boldsymbol{z} > r_k\boldsymbol{w}^T\boldsymbol{\eta} - \epsilon\|\boldsymbol{w}\|)$$
(12)

Since $\boldsymbol{z} \sim \mathcal{N}(0, \mathbf{I})$, we have $-y\boldsymbol{w}^T\boldsymbol{z} \sim \mathcal{N}(0, (-y\boldsymbol{w}^T)^T(-y\boldsymbol{w}^T)) = \mathcal{N}(0, \boldsymbol{w}^T\boldsymbol{w})$. Furthermore $\frac{-y\boldsymbol{w}^T\boldsymbol{z}}{\|\boldsymbol{w}\|} \sim \mathcal{N}(0, 1)$, and we can further simplify $\widehat{\mathcal{R}}_k(\epsilon)$ as follows:

$$\widehat{\mathcal{R}}_k(\epsilon) = \mathbb{P}(\frac{-y\boldsymbol{w}^T\boldsymbol{z}}{\|\boldsymbol{w}\|} > \frac{r_k\boldsymbol{w}^T\boldsymbol{\eta}}{\|\boldsymbol{w}\|} - \epsilon) = \Phi(\frac{r_k\boldsymbol{w}^T\boldsymbol{\eta}}{\|\boldsymbol{w}\|} - \epsilon)$$
(13)

$\square$

With Lemma 2, we can straightforwardly calculate the robust error for all components of the GMM model defined in (3):

$$\widehat{\mathcal{R}}(\epsilon) = \sum_{k=1}^{K} p_k\Phi(\frac{r_k\boldsymbol{w}^T\boldsymbol{\eta}}{\|\boldsymbol{w}\|} - \epsilon)$$
(14)

On the other hand, Theorem 1 indicates the parameter $\boldsymbol{w}$ will converge to the $l_2$ max margin. However, for arbitrary training set, we do not have the closed form of $\boldsymbol{w}$, which is a barrier for the further analysis. Nevertheless, results from (Wang & Thrampoulidis, 2020) indicates in the over-parameterization regime, the parameter $\boldsymbol{w}$ will converge to min-norm interpolation of the data with high probability.

**Lemma 3.** *(Directly from Theorem 1 in (Wang & Thrampoulidis, 2020)) Assume $n$ training instances drawn from the $l$-th mode of the described distribution in (3) and each of them is a $m$-dimensional vector. If $\frac{m}{n \log n}$ is sufficiently large[4], then the $l_2$ max margin vector in Equation (2) will be the same as the solution of the min-norm interpolation described below with probability at least $(1 - O(\frac{1}{n}))$.*

$$\bar{\boldsymbol{w}} = \arg\min_{\boldsymbol{w}} \|\boldsymbol{w}\| \ \ s.t. \ \forall i \in \{1, 2, ..., n\}, \ y_i = \boldsymbol{w}^T\boldsymbol{x}_i$$
(15)

Since the min-norm interpolation has a closed solution $\bar{\boldsymbol{w}} = \mathbf{X}^T(\mathbf{X}\mathbf{X}^T)^{-1}\boldsymbol{y}$, Lemma 3 will greatly facilitate the calculation of $\mathbb{R}(\boldsymbol{w})$ in Theorem 2. To simplify the notation, we first define the following variables.

$$\mathbf{U} = \mathbf{Q}\mathbf{Q}^T, \ \boldsymbol{d} = \mathbf{Q}\boldsymbol{\eta}, \ s = \boldsymbol{y}^T\mathbf{U}^{-1}\boldsymbol{y}, \ t = \boldsymbol{d}\mathbf{U}^{-1}\boldsymbol{d}, \ v = \boldsymbol{y}^T\mathbf{U}^{-1}\boldsymbol{d}$$
(16)

The proof of Theorem 2 is then presented below.

*Proof.* Based on (14), the key is to simplify the term $\frac{\boldsymbol{w}^T\boldsymbol{\eta}}{\|\boldsymbol{w}\|}$, let's denote it by $A$, then we have:

$$A^2 = \frac{\boldsymbol{\eta}^T\boldsymbol{w}\boldsymbol{w}^T\boldsymbol{\eta}}{\boldsymbol{w}^T\boldsymbol{w}} = \frac{(\boldsymbol{y}^T(\mathbf{X}\mathbf{X}^T)^{-1}\mathbf{X}\boldsymbol{\eta})^2}{\boldsymbol{y}^T(\mathbf{X}\mathbf{X}^T)^{-1}\boldsymbol{y}}$$
(17)

The key challenge here is to calculate the term $(\mathbf{X}\mathbf{X}^T)^{-1}$ where $\mathbf{X} = r_l\boldsymbol{y}\boldsymbol{\eta}^T + Q$. Here we utilize Lemma 3 of (Wang & Thrampoulidis, 2020) and Woodbury identity (Horn & Johnson, 2012), we have:

$$\boldsymbol{y}^T(\mathbf{X}\mathbf{X})^{-1} = \boldsymbol{y}^T\mathbf{U}^{-1} - \frac{(r_l^2 s\|\boldsymbol{\eta}\|^2 + r_l^2 v^2 + r_l v - r_l^2 st)\boldsymbol{y}^T + r_l s\boldsymbol{d}^T}{r_l^2 s(\|\boldsymbol{\eta}\|^2 - t) + (r_l v + 1)^2}\mathbf{U}^{-1}$$
(18)

---

[4]Specifically, $m$ and $n$ need to satisfy $m > 10n\log n + n - 1$ and $m > Cnr_l\sqrt{\log 2n}\|\boldsymbol{\eta}\|$. The constant $C$ is derived in the proof of Theorem 1 in (Wang & Thrampoulidis, 2020).

Here, $s$, $t$, $v$, $\mathbf{U}$ and $\boldsymbol{d}$ are defined in Equation (16). The scalar divisor comes from the matrix inverse calculation. Base of Equation (18), we can then calculate $\boldsymbol{y}^T(\mathbf{XX}^T)^{-1}\boldsymbol{y}$ and $\boldsymbol{y}^T(\mathbf{XX}^T)^{-1}\mathbf{X}\boldsymbol{\eta}$.

$$
\begin{aligned}
\boldsymbol{y}^T(\mathbf{XX}^T)^{-1}\boldsymbol{y} &= s - \frac{(r_l^2 s\|\boldsymbol{\eta}\|^2 + r_l^2 v^2 + r_l v - r_l^2 st)s + r_l sv}{r_l^2 s(\|\boldsymbol{\eta}\|^2 - t) + (r_l v + 1)^2} \\
&= \frac{s}{r_l^2 s(\|\boldsymbol{\eta}\|^2 - t) + (r_l v + 1)^2}
\end{aligned}
\tag{19}
$$

$$
\begin{aligned}
\boldsymbol{y}^T(\mathbf{XX}^T)^{-1}\mathbf{X}\boldsymbol{\eta} &= \boldsymbol{y}^T(\mathbf{XX}^T)^{-1}(r_l \boldsymbol{y}\boldsymbol{\eta}^T + Q)\boldsymbol{\eta} \\
&= r_l\|\boldsymbol{\eta}\|^2 \boldsymbol{y}^T(\mathbf{XX}^T)^{-1}\boldsymbol{y} + \boldsymbol{y}^T(\mathbf{XX}^T)^{-1}\boldsymbol{d} \\
&= \frac{r_l s(\|\boldsymbol{\eta}\|^2 - t) + r_l v^2 + v}{r_l^2 s(\|\boldsymbol{\eta}\|^2 - t) + (r_l v + 1)^2}
\end{aligned}
\tag{20}
$$

Plug Equation (19) and (20) into (17), we have:

$$
\begin{aligned}
A^2 &= \frac{\left(r_l s(\|\boldsymbol{\eta}\|^2 - t) + r_l v^2 + v\right)^2}{s\left(r_l^2 s(\|\boldsymbol{\eta}\|^2 - t) + (r_l v + 1)^2\right)} \\
&= \frac{s(\|\boldsymbol{\eta}\|^2 - t) + v^2}{s} - \frac{\|\boldsymbol{\eta}\|^2 - t}{r_l^2 s(\|\boldsymbol{\eta}\|^2 - t) + (r_l v + 1)^2} \\
&= \frac{s(\|\boldsymbol{\eta}\|^2 - t) + v^2}{s} - \frac{1}{\left(\frac{s(\|\boldsymbol{\eta}\|^2 - t) + v^2}{\|\boldsymbol{\eta}\|^2 - t}\right)r_l^2 + \frac{2v}{\|\boldsymbol{\eta}\|^2 - t}r_l + \frac{1}{\|\boldsymbol{\eta}\|^2 - t}}
\end{aligned}
\tag{21}
$$

Plug (21) into (14), we then obtain the robust error on all components of the GMM defined in (3):

$$
\mathcal{R}(r_l, \epsilon) = \sum_{k=1}^{K} p_k \Phi\left(r_k g(r_l) - \epsilon\right), \ g(r_l) = (C_1 - \frac{1}{C_2 r_l^2 + C_3})^{\frac{1}{2}}
\tag{22}
$$

$$
C_1 = \frac{s(\|\boldsymbol{\eta}\|^2 - t) + v^2}{s}, \ C_2 = \frac{s(\|\boldsymbol{\eta}\|^2 - t) + v^2}{\|\boldsymbol{\eta}\|^2 - t}, \ C_3 = \frac{2v}{\|\boldsymbol{\eta}\|^2 - t}r_l + \frac{1}{\|\boldsymbol{\eta}\|^2 - t}.
$$

We study the sign of $C_1$ and $C_2$. Consider $\mathbf{U} = \mathbf{QQ}^T$ is a positive semidefinite matrix, so $s = \boldsymbol{y}\mathbf{U}^{-1}\boldsymbol{y}^T \geq 0$. In addition, we have $\|\boldsymbol{\eta}\|^2 - t = \boldsymbol{\eta}^T\left(\mathbf{I} - (\mathbf{QQ}^T)^{-1}\right)\boldsymbol{\eta}$. Since $\mathbf{I} - (\mathbf{QQ}^T)^{-1} = (\mathbf{I} - (\mathbf{QQ}^T)^{-1})^T(\mathbf{I} - (\mathbf{QQ}^T)^{-1})$ is a positive semidefinite matrix, we can obtain $\mathbf{I} - (\mathbf{QQ}^T)^{-1}$ is also a positive semidefinite matrix. As a result, $C_1$ and $C_2$ are both non-negative.

$\square$

### B.3 Proof of Corollary 1

To prove Corollary 1, we first prove the following lemma:

**Lemma 4.** *Under the condition of Theorem 2 and $\mathcal{R}$ in Equation (4), $\frac{\partial \mathcal{R}(r_l, \epsilon)}{\partial r_l}$ is negative and monotonically decreases with $\epsilon$.*

*Proof.* Based on Equation (22), we have:

$$
\frac{\partial \mathcal{R}(r_l, \epsilon)}{\partial r_l} = \sum_{k=1}^{K} p_k \Phi'(r_k g(r_l) - \epsilon)\frac{\partial g(r_l)}{\partial r_l}
\tag{23}
$$

Since the training data is separable, we have $\forall k, r_k \boldsymbol{w}^T\boldsymbol{\eta} - \epsilon\|\boldsymbol{w}\| > 0$, which is equivalent to the following:

$$
\forall k, r_k g(r_l) - \epsilon > 0
\tag{24}
$$

First, $p_k$ is a positive number by definition. Consider function $\Phi(x)$ monotonically decrease with $x$ and is convex when $x > 0$, so $\forall k$, $\Phi'(r_k g(r_l) - \epsilon)$ is negative and decreases with $\epsilon$. In addition, $g(r_l)$ increases with $r_l$ and is independent on $\epsilon$, so $\frac{\partial g(r_l)}{\partial r_l}$ can be considered as a positive constant. Therefore, $\frac{\partial \mathcal{R}(r_l, \epsilon)}{\partial r_l}$ is negative and monotonically decreases with $\epsilon$.

$\square$

Now, we are ready to prove Corollary 1:

*Proof.* We subtract the left hand side from the right hand side in the inequality of Corollary 1:

$$
\begin{aligned}
[\mathcal{R}(r_j, \epsilon_1) - \mathcal{R}(r_i, \epsilon_1)] - [\mathcal{R}(r_j, \epsilon_2) - \mathcal{R}(r_i, \epsilon_2)] &= \int_{r_i}^{r_j} \frac{\partial \mathcal{R}(r_l, \epsilon_1)}{\partial r_l} dr_l - \int_{r_i}^{r_j} \frac{\partial \mathcal{R}(r_l, \epsilon_2)}{\partial r_l} dr_l \\
&= \int_{r_i}^{r_j} \left[ \frac{\partial \mathcal{R}(r_l, \epsilon_1)}{\partial r_l} - \frac{\partial \mathcal{R}(r_l, \epsilon_2)}{\partial r_l} \right] dr_l \\
&> 0
\end{aligned}
\tag{25}
$$

The last inequality is based on $r_j > r_i$, $\epsilon_2 > \epsilon_1$, Lemma 4, which indicates $\left[ \frac{\partial \mathcal{R}(r_l, \epsilon_1)}{\partial r_l} - \frac{\partial \mathcal{R}(r_l, \epsilon_2)}{\partial r_l} \right]$ is always positive. We reorganize (25) and obtain $\mathcal{R}(r_i, \epsilon_1) - \mathcal{R}(r_j, \epsilon_1) < \mathcal{R}(r_i, \epsilon_2) - \mathcal{R}(r_j, \epsilon_2)$.

$\square$

### B.4 PROOF OF THEOREM 3

We start with the following lemma.

**Lemma 5.** *Given the assumptions of Theorem 3, we define $g(\boldsymbol{x}) = \mathbb{E}(y|\boldsymbol{x})$, $z(\boldsymbol{x}) = y - g(\boldsymbol{x})$ and consider $\gamma = \sigma_l^2 + h^2(C, \epsilon) - C$, then the following inequality holds.*

$$
\forall a \in (0, 1), \mathbb{P}(\exists f_{\boldsymbol{w}} \in \mathcal{F} : \frac{1}{n} \sum_{i=1}^{n} (y_i - f_{\boldsymbol{w}}(\boldsymbol{x}_i'))^2 \leq C)
$$
$$
\leq 2e^{-\frac{na^2\gamma^2}{8}} + \mathbb{P}(\exists f_{\boldsymbol{w}} \in \mathcal{F} : \frac{1}{n} \sum_{i=1}^{n} f_{\boldsymbol{w}}(\boldsymbol{x}_i) z(\boldsymbol{x}_i) \geq \frac{1}{2}(1 - 3a)\gamma)
\tag{26}
$$

*Proof.* Given the definition of $h(C, \epsilon)$, we have:

$$
\begin{aligned}
(y_i - f_{\boldsymbol{w}}(\boldsymbol{x}_i'))^2 &= [(y_i - f_{\boldsymbol{w}}(\boldsymbol{x}_i)) + (f_{\boldsymbol{w}}(\boldsymbol{x}_i) - f_{\boldsymbol{w}}(\boldsymbol{x}_i'))]^2 \\
&\geq (y_i - f_{\boldsymbol{w}}(\boldsymbol{x}_i))^2 + (f_{\boldsymbol{w}}(\boldsymbol{x}_i) - f_{\boldsymbol{w}}(\boldsymbol{x}_i'))^2 \\
&\geq (y_i - f_{\boldsymbol{w}}(\boldsymbol{x}_i))^2 + h^2(C, \epsilon)
\end{aligned}
\tag{27}
$$

For the first inequality, $\boldsymbol{x}_i'$ is the adversarial example which tries to maximize the loss objective, $y_i \in \{-1, +1\}$ and the range of $f_{\boldsymbol{w}}$ is $[-1, +1]$, so $\langle y_i - f_{\boldsymbol{w}}(\boldsymbol{x}_i), f_{\boldsymbol{w}}(\boldsymbol{x}_i) - f_{\boldsymbol{w}}(\boldsymbol{x}_i') \rangle \geq 0$. The second inequality is based on the definition of $h^2(C, \epsilon)$ in Definition 1. As a result, we can simplify the left hand side of (26) as follows:

$$
\mathbb{P}(\exists f_{\boldsymbol{w}} \in \mathcal{F} : \frac{1}{n} \sum_{i=1}^{n} (y_i - f_{\boldsymbol{w}}(\boldsymbol{x}_i'))^2 \leq C) \leq \mathbb{P}(\exists f_{\boldsymbol{w}} \in \mathcal{F} : \frac{1}{n} \sum_{i=1}^{n} (y_i - f_{\boldsymbol{w}}(\boldsymbol{x}_i))^2 \leq C - h^2(C, \epsilon))
\tag{28}
$$

We consider the sequence $\{z(\boldsymbol{x}_i)\}_{i=1}^n$, it is i.i.d with $\mathbb{E}_{\mu_l}(z(\boldsymbol{x})^2) = \mathbb{E}_{\mu_l}[Var(y|\boldsymbol{x})] = \sigma_l^2$. Since the range of the prediction is $[-1, +1]$, so $z^2(\boldsymbol{x}) \in [0, 4]$. Then, we have the following inequality by Hoeffding's inequality (Hoeffding, 1994).

$$\forall a \in (0, 1), \mathbb{P}(\frac{1}{n}\sum_{i=1}^n z^2(\boldsymbol{x}_i) \leq \sigma_l^2 - a\gamma) \leq e^{-\frac{na^2\gamma^2}{8}} \tag{29}$$

Similarly, we consider the sequence $\{z(\boldsymbol{x}_i)g(\boldsymbol{x}_i)\}_{i=1}^n$, the following inequality holds based on the Hoeffding's inequality and the fact $\mathbb{E}(z(\boldsymbol{x})g(\boldsymbol{x})) = 0$, $z(\boldsymbol{x})g(\boldsymbol{x}) \in [-2, +2]$.

$$\forall a \in (0, 1), \mathbb{P}(\frac{1}{n}\sum_{i=1}^n z(\boldsymbol{x}_i)g(\boldsymbol{x}_i) \leq a\gamma) \leq e^{-\frac{na^2\gamma^2}{8}} \tag{30}$$

Now we study the right hand side of (28):

$$\frac{1}{n}\sum_{i=1}^n (y_i - f_{\boldsymbol{w}}(\boldsymbol{x}_i))^2 = \frac{1}{n}\sum_{i=1}^n \left(z^2(\boldsymbol{x}_i) + (g(\boldsymbol{x}_i) - f_{\boldsymbol{w}}(\boldsymbol{x}_i))^2 + 2z(\boldsymbol{x}_i)(g(\boldsymbol{x}_i) - f_{\boldsymbol{w}}(\boldsymbol{x}_i))\right)$$
$$\geq \frac{1}{n}\sum_{i=1}^n \left(z^2(\boldsymbol{x}_i) + 2z(\boldsymbol{x}_i)g(\boldsymbol{x}_i) - 2z(\boldsymbol{x}_i)f_{\boldsymbol{w}}(\boldsymbol{x}_i)\right) \tag{31}$$

Consider the following reasoning:

$$\begin{cases} \dfrac{1}{n}\sum_{i=1}^n (y_i - f_{\boldsymbol{w}}(\boldsymbol{x}_i))^2 \leq C - h^2(C, \epsilon) = \sigma_l^2 - \gamma \\[2ex] \dfrac{1}{n}\sum_{i=1}^n z^2(\boldsymbol{x}_i) \geq \sigma_l^2 - a\gamma \\[2ex] \dfrac{1}{n}\sum_{i=1}^n z(\boldsymbol{x}_i)g(\boldsymbol{x}_i) \geq -a\gamma \end{cases} \implies \frac{1}{n}\sum_{i=1}^n z(\boldsymbol{x}_i)f_{\boldsymbol{w}}(\boldsymbol{x}_i) \geq \frac{1}{2}(1 - 3a)\gamma \tag{32}$$

As a result, we have:

$$\mathbb{P}(\exists f_{\boldsymbol{w}} \in \mathcal{F} : \frac{1}{n}\sum_{i=1}^n (y_i - f_{\boldsymbol{w}}(\boldsymbol{x}_i) \leq C - h^2(C, \epsilon)))$$
$$\leq \mathbb{P}(\exists f_{\boldsymbol{w}} \in \mathcal{F} : \frac{1}{n}\sum_{i=1}^n z^2(\boldsymbol{x}_i) \leq \sigma_l^2 - a\gamma) + \mathbb{P}(\exists f_{\boldsymbol{w}} \in \mathcal{F} : \frac{1}{n}\sum_{i=1}^n z(\boldsymbol{x}_i)g(\boldsymbol{x}_i) \geq -a\gamma) +$$
$$\mathbb{P}(\exists f_{\boldsymbol{w}} \in \mathcal{F} : \frac{1}{n}\sum_{i=1}^n z(\boldsymbol{x}_i)f_{\boldsymbol{w}}(\boldsymbol{x}_i) \geq \frac{1}{2}(1 - 3a)\gamma)$$
$$\leq 2e^{-\frac{na^2\gamma^2}{8}} + \mathbb{P}(\exists f_{\boldsymbol{w}} \in \mathcal{F} : \frac{1}{n}\sum_{i=1}^n z(\boldsymbol{x}_i)f_{\boldsymbol{w}}(\boldsymbol{x}_i) \geq \frac{1}{2}(1 - 3a)\gamma) \tag{33}$$

The first inequality is based on the reasoning of (32). The second inequality is based on (29) and (30).

Based on the inequality (28) and (33), we conclude the proof.

$\square$

To further simplify the right hand side of (26), $\mathbb{P}(\exists f_{\boldsymbol{w}} \in \mathcal{F} : \frac{1}{n}\sum_{i=1}^{n} z(\boldsymbol{x}_i)f_{\boldsymbol{w}}(\boldsymbol{x}_i) \geq \frac{1}{2}(1-3a)\gamma)$ needs to be bounded, and this is solved by the following lemma.

**Lemma 6.** *Given the assumptions of Theorem 3 and the definition of $g(\boldsymbol{x})$, $z(\boldsymbol{x})$ in Lemma 5, then the following inequality holds.*

$$\forall a \in (0,1), a_1 > 0, a_2 > 0 \text{ and } a_1 + a_2 = \frac{1}{2}(1-3a),$$

$$\mathbb{P}(\exists f_{\boldsymbol{w}} \in \mathcal{F} : \frac{1}{n}\sum_{i=1}^{n} z(\boldsymbol{x}_i)f_{\boldsymbol{w}}(\boldsymbol{x}_i) \geq \frac{1}{2}(1-3a)\gamma) \leq 2|\mathcal{F}|e^{-\frac{nm}{144cL^2}a_1^2\gamma^2} + 2e^{-\frac{n}{8}a_2^2\gamma^2} \tag{34}$$

*Proof.* We recall that the data points $\{\boldsymbol{x}_i, y_i\}_{i=1}^{n}$ are sampled from the distribution $\mu_l$, which is $c$-isoperimetric. For any $L$-Lipschitz function $f$, we have:

$$\forall t, \mathbb{P}[|f_{\boldsymbol{w}}(\boldsymbol{x}) - \mathbb{E}_{\mu_l}(f_{\boldsymbol{w}})| \geq t] \leq 2e^{-\frac{mt^2}{2cL^2}} \tag{35}$$

Since $z(\boldsymbol{x}) = y - g(\boldsymbol{x}) \in [-2, +2]$, we can then bound $\mathbb{P}[z(\boldsymbol{x})(f_{\boldsymbol{w}}(\boldsymbol{x}) - \mathbb{E}_{\mu_l}(f_{\boldsymbol{w}})) \geq t]$:

$$\forall t, \mathbb{P}[z(\boldsymbol{x})(f_{\boldsymbol{w}}(\boldsymbol{x}) - \mathbb{E}_{\mu_l}(f_{\boldsymbol{w}})) \geq t] \leq \mathbb{P}[|z(\boldsymbol{x})(f_{\boldsymbol{w}}(\boldsymbol{x}) - \mathbb{E}_{\mu_l}(f_{\boldsymbol{w}}))| \geq t]$$

$$\leq \mathbb{P}[|(f_{\boldsymbol{w}}(\boldsymbol{x}) - \mathbb{E}_{\mu_l}(f_{\boldsymbol{w}}))| \geq \frac{t}{2}] \leq 2e^{-\frac{mt^2}{8cL^2}} \tag{36}$$

Here we utilize the proposition in (Vershynin, 2018; Van Handel, 2014)[5], which claims *if $\{X_i\}_{i=1}^{n}$ are independent variables and all $C$-subgaussian, then $\frac{1}{\sqrt{n}}\sum_{i=1}^{n} X_i$ is $18C$-subgaussian.* Therefore, we have:

$$\forall t, \mathbb{P}[\frac{1}{\sqrt{n}}\sum_{i=1}^{n} z(\boldsymbol{x}_i)(f_{\boldsymbol{w}}(\boldsymbol{x}_i) - \mathbb{E}_{\mu_l}(f_{\boldsymbol{w}})) \geq t] \leq 2e^{-\frac{mt^2}{144cL^2}} \tag{37}$$

Let $t = a_1\gamma\sqrt{n}$, then we have:

$$\mathbb{P}[\frac{1}{n}\sum_{i=1}^{n} z(\boldsymbol{x}_i)(f_{\boldsymbol{w}}(\boldsymbol{x}_i) - \mathbb{E}_{\mu_l}(f)) \geq a_1\gamma] \leq 2e^{-\frac{nm}{144cL^2}a_1^2\gamma^2} \tag{38}$$

In addition, we can bound $\mathbb{P}[\frac{1}{n}\sum_{i=1}^{n} z(\boldsymbol{x}_i)\mathbb{E}_{\mu_l}(f_{\boldsymbol{w}}) \geq a_2\gamma]$ by:

$$\mathbb{P}[\exists f_{\boldsymbol{w}} \in \mathcal{F} : \frac{1}{n}\sum_{i=1}^{n} z(\boldsymbol{x}_i)\mathbb{E}_{\mu_l}(f_{\boldsymbol{w}}) \geq a_2\gamma] \leq \mathbb{P}[\frac{1}{n}\sum_{i=1}^{n}|z(\boldsymbol{x}_i)| \geq a_2\gamma] \leq 2e^{-\frac{n}{8}a_2^2\gamma^2} \tag{39}$$

The first inequality is based on the fact $\mathbb{E}_{\mu_l}(f_{\boldsymbol{w}}) \in [-1, +1]$; the second inequality is based on Hoeffding's inequality.

Now, we are ready to bound the probability $\mathbb{P}(\exists f_{\boldsymbol{w}} \in \mathcal{F} : \frac{1}{n}\sum_{i=1}^{n} z(\boldsymbol{x}_i)f_{\boldsymbol{w}}(\boldsymbol{x}_i) \geq \frac{1}{2}(1-3a)\gamma)$.

$$\mathbb{P}(\exists f_{\boldsymbol{w}} \in \mathcal{F} : \frac{1}{n}\sum_{i=1}^{n} z(\boldsymbol{x}_i)f_{\boldsymbol{w}}(\boldsymbol{x}_i) \geq \frac{1}{2}(1-3a)\gamma)$$

$$\leq \mathbb{P}[\exists f_{\boldsymbol{w}} \in \mathcal{F} : \frac{1}{n}\sum_{i=1}^{n} z(\boldsymbol{x}_i)(f_{\boldsymbol{w}}(\boldsymbol{x}_i) - \mathbb{E}_{\mu_l}(f)) \geq a_1\gamma] + \mathbb{P}[\exists f_{\boldsymbol{w}} \in \mathcal{F} : \frac{1}{n}\sum_{i=1}^{n} z(\boldsymbol{x}_i)\mathbb{E}_{\mu_l}(f_{\boldsymbol{w}}) \geq a_2\gamma]$$

$$\leq 2|\mathcal{F}|e^{-\frac{nm}{144cL^2}a_1^2\gamma^2} + 2e^{-\frac{n}{8}a_2^2\gamma^2} \tag{40}$$

The first inequality is based on the fact $a_1 + a_2 = \frac{1}{2}(1-3a)$; the second inequality is based on the Boole's inequality (Boole, 1847), inequality (38) and (39).

$\square$

---

[5]Proposition 2.6.1 in (Vershynin, 2018) and Exercise 3.1 in (Van Handel, 2014)

To simplify the constant notation, we let $a = \frac{1}{8}$, $a_1 = \frac{3}{16}$ and $a_2 = \frac{1}{8}$. We plug this into the inequality (26) and (34), then:

$$\mathbb{P}(\exists f_{\boldsymbol{w}} \in \mathcal{F} : \frac{1}{n} \sum_{i=1}^{n} (y_i - f_{\boldsymbol{w}}(\boldsymbol{x}_i'))^2 \leq C) \leq 4e^{-\frac{n\gamma^2}{2^9}} + 2|\mathcal{F}|e^{-\frac{nm\gamma^2}{2^{12}cL^2}} \tag{41}$$

Now we turn to the proof of Theorem 3.

*Proof.* We let $\mathcal{F}_L = \{f_{\boldsymbol{w}} | \boldsymbol{w} \in \mathcal{W}, Lip(f_{\boldsymbol{w}}) \leq L\}$, $\mathcal{F}_\gamma = \{f_{\boldsymbol{w}} | \boldsymbol{w} \in \mathcal{W}, \boldsymbol{w} = \frac{\gamma}{4J} \odot \boldsymbol{z}, \boldsymbol{z} \in \mathbb{Z}^b\}$ and $\mathcal{F}_{\gamma,L} = \mathcal{F}_\gamma \cap \mathcal{F}_L$. Correspondingly, we let $\mathcal{W}_L = \{\boldsymbol{w} | \boldsymbol{w} \in \mathcal{W}, Lip(f_{\boldsymbol{w}}) \leq L\}$, $\mathcal{W}_\gamma = \{\boldsymbol{w} | \boldsymbol{w} \in \mathcal{W}, \boldsymbol{w} = \frac{\gamma}{4J} \odot \boldsymbol{z}, \boldsymbol{z} \in \mathbb{Z}^b\}$ and $\mathcal{W}_{\gamma,L} = \mathcal{W}_\gamma \cap \mathcal{W}_L$. Because the diameter of $\mathcal{W}$ is $W$, we have $|\mathcal{F}_{\gamma,L}| \leq |\mathcal{F}_\gamma| \leq \left(\frac{4WJ}{\gamma}\right)^b$. Here, $\odot$ means the element-wise multiplication.

Note that the inequality (41) is valid for any values of $C$ as long as it satisfies $\gamma \geq 0$. Based on this, we apply the substitution $\begin{cases} C \leftarrow C + \frac{1}{2}\gamma \\ \gamma \leftarrow \frac{1}{2}\gamma \end{cases}$, then:

$$\mathbb{P}(\exists f_{\boldsymbol{w}} \in \mathcal{F}_{\gamma,L} : \frac{1}{n} \sum_{i=1}^{n} (y_i - f_{\boldsymbol{w}}(\boldsymbol{x}_i'))^2 \leq C + \frac{1}{2}\gamma) \leq 4e^{-\frac{n\gamma^2}{2^{11}}} + 2|\mathcal{F}|e^{-\frac{nm\gamma^2}{2^{14}cL^2}} \tag{42}$$

$$\leq 4e^{-\frac{n\gamma^2}{2^{11}}} + 2e^{b\log(\frac{4WJ}{\gamma}) - \frac{nm\gamma^2}{2^{14}cL^2}}$$

Based on the definition of $\mathcal{W}_{\gamma,L}$, we can conclude that $\forall \boldsymbol{w}_1 \in \mathcal{W}_L, \exists \boldsymbol{w}_2 \in \mathcal{W}_{\gamma,L} \ s.t. \|\boldsymbol{w}_1 - \boldsymbol{w}_2\|_\infty \leq \frac{\gamma}{8J}$. Therefore, $\forall f_{\boldsymbol{w}_1} \in \mathcal{F}_L, \exists f_{\boldsymbol{w}_2} \in \mathcal{F}_{\gamma,L} s.t. \|f_{\boldsymbol{w}_1} - f_{\boldsymbol{w}_2}\|_\infty \leq \frac{\gamma}{8}$. Let choose such $f_{\boldsymbol{w}_2} \in \mathcal{F}_{\gamma,L}$ given an arbitrary $f_{\boldsymbol{w}_1} \in \mathcal{F}_L$, then:

$$(y - f_{\boldsymbol{w}_1}(\boldsymbol{x}))^2 = (y - f_{\boldsymbol{w}_2}(\boldsymbol{x}))^2 + (2y - f_{\boldsymbol{w}_1}(\boldsymbol{x}) - f_{\boldsymbol{w}_2}(\boldsymbol{x}))(f_{\boldsymbol{w}_2}(\boldsymbol{x}) - f_{\boldsymbol{w}_1}(\boldsymbol{x}))$$
$$\geq (y - f_{\boldsymbol{w}_2}(\boldsymbol{x}))^2 - \frac{\gamma}{8}|(2y - f_{\boldsymbol{w}_1}(\boldsymbol{x}) - f_{\boldsymbol{w}_2}(\boldsymbol{x}))| \tag{43}$$
$$\geq (y - f_{\boldsymbol{w}_2}(\boldsymbol{x}))^2 - \frac{\gamma}{2}$$

The first inequality in (43) is based on Hölder's inequality; the second inequality is based on $y \in \{-1, +1\}$ and the range of $\forall f_{\boldsymbol{w}} \in \mathcal{F}$ is $[-1, +1]$.

We combine (41) with (43), then:

$$\mathbb{P}(\exists f_{\boldsymbol{w}} \in \mathcal{F}_L : \frac{1}{n} \sum_{i=1}^{n} (y_i - f_{\boldsymbol{w}}(\boldsymbol{x}_i'))^2 \leq C) \leq \mathbb{P}(\exists f_{\boldsymbol{w}} \in \mathcal{F}_{\gamma,L} : \frac{1}{n} \sum_{i=1}^{n} (y_i - f_{\boldsymbol{w}}(\boldsymbol{x}_i'))^2 \leq C + \frac{1}{2}\gamma)$$

$$\leq 4e^{-\frac{n\gamma^2}{2^{11}}} + 2e^{b\log(\frac{4WJ}{\gamma}) - \frac{nm\gamma^2}{2^{14}cL^2}} \tag{44}$$

Note that $\mathcal{F}_L$ is the set of functions in $\mathcal{F}$ whose Lipschitz constant is no larger than $L$. We set the right hand side of (44) to be $\delta$ and then get $L = \frac{\gamma}{2^7} \sqrt{\frac{nm}{c(b\log(4WJ\gamma^{-1}) - \log(\delta/2 - 2e^{-2^{-11}n\gamma^2}))}}$. This concludes the proof.

$\square$

## C  EXPERIMENTAL SETTINGS

### C.1  GENERAL SETTINGS

The ResNet-18 (RN18) architecture is same as the one in (Wong et al., 2020); the WideResNet-34 (WRN34) architecture is same as the one in (Madry et al., 2018). Unless specified, the $l_\infty$ adversarial budget used for CIFAR10 dataset (Krizhevsky et al., 2009) [6] is $8/255$ and for SVHN dataset (Netzer et al., 2011) [7] is $0.02$. In PGD adversarial training, the step size is $2/255$ for CIFAR10 and $0.005$ for SVHN; PGD is run for $10$ iterations for both datasets. For adversarial attacks using a different adversarial budget, the step size is always $1/4$ of the adversarial budget's size, and we always run it for $10$ iterations. To comprehensively and reliably evaluate the robustness of the model, we use AutoAttack (Croce & Hein, 2020b), which is an ensemble of $4$ different attacks: AutoPGD on cross entropy, AutoPGD on difference of logits ratio, fast adaptive boundary (FAB) attack (Croce & Hein, 2020a) and square attack (Andriushchenko et al., 2020). Unless specified, we use stochastic gradient descent (SGD) with a momentum to optimize the model parameters, we also use weight decay whose factor is $0.0005$. Unless specified, the momentum factor is $0.9$, the learning rate starts with $0.1$ and is divided by $10$ in the $1/2$ and $3/4$ of the whole training duration. The size of the mini-batch is always $128$.

We run the experiments on a machine with 4 NVIDIA TITAN XP GPUs. It takes about 6 hours to adversarially train a RN18 model for 200 epochs, and a whole day to adversarially train a WRN34 model for 200 epochs.

### C.2  SETTINGS OF EXPERIMENTS IN SECTION 6

---

**Algorithm 1:** One epoch of the accelerated adversarial training we use in Section 6.1.

---

**Input:** training data $\mathcal{D}$, model $f$, batch size $B$, PGD step size $\alpha$, adversarial budget $\mathcal{S}^{(p)}(\epsilon)$, coefficient $\rho$, $\beta$.
**for** Sample a mini-batch $\{\boldsymbol{x}_i, y_i\}_{i=1}^B \sim \mathcal{D}$ **do**
    $\forall i$, obtain the initial perturbation $\Delta_i$ as in (Zheng et al., 2020).
    $\forall i$, one step PGD update: $\Delta_i \leftarrow \Pi_{\mathcal{S}^{(p)}(\epsilon)}[\Delta_i + \alpha sign(\nabla_{\Delta_i}\mathcal{L}_\theta(\boldsymbol{x}_i + \Delta_i, y_i))]$.
    $\forall i$, update the cached adversarial perturbation $\Delta_i$ as in (Zheng et al., 2020).
    **if** use reweight **then**
        $\forall i$, weight $w_i = softmax[f(\boldsymbol{x}_i + \Delta_i)]_{y_i}$
    **else**
        $\forall i$, weight $w_i = 1$
    **end if**
    $\forall i$, query the adaptive target $\tilde{\boldsymbol{t}}_i$ and update: $\tilde{\boldsymbol{t}}_i \leftarrow \rho\tilde{\boldsymbol{t}}_i + (1-\rho)softmax[f(\boldsymbol{x}_i + \Delta_i)]$.
    $\forall i$, the final adaptive target $\boldsymbol{t}_i = \beta\mathbf{1}_{y_i} + (1-\beta)\tilde{\boldsymbol{t}}_i$
    Calculate the loss $\frac{1}{\sum_i^B w_i}\sum_i^B w_i\mathcal{L}_\theta(\boldsymbol{x}_i + \Delta_i, \boldsymbol{t}_i)$ and update the parameters.
**end for**

---

**Fast Adversarial Training** Our experiment on fast adversarial training is on CIFAR10 and $\epsilon = 8/255$. The pseudocode of our method is demonstrated as Algorithm 1. We use ATTA (Zheng et al., 2020) to initialize the perturbation of each training instance. The step size $\alpha$ of the perturbation update is $4/255$, same as (Zheng et al., 2020). The average coefficient $\rho$ and $\beta$ is $0.9$ and $0.1$ unless explicitly stated. Our learning rate scheduler also follows (Zheng et al., 2020): we train the model for 38 epochs, the learning rate is $0.1$ on the first 30 epochs, it decays to $0.01$ in the next 6 epochs and further decays to $0.001$ in the last 2 epochs. When we use adaptive targets, the first 5 epochs are the warmup period in which we use fixed targets. Since the goal here is to accelerate adversarial training, we do not use a validation set to do model selection as in (Rice et al., 2020). We use the standard data augmentation on CIFAR10: random crop and random horizontal flip.

---

[6]Data available for download on https://www.cs.toronto.edu/ kriz/cifar.html. MIT license. Free to use.
[7]Data available for download on http://ufldl.stanford.edu/housenumbers/. Free for non-commercial use.

**Adversarial Finetuning with Additional Data** For CIFAR10, we use 500000 images from 80 Million Tiny Images dataset (Torralba et al., 2008) with pseudo labels in (Carmon et al., 2019) [8]. For SVHN, we use the extra held-out set provided by SVHN itself, which contains 531131 somewhat less difficult samples. When we construct a mini-batch, half of its instances are sampled from the original training set and the other half are sampled from the additional data. The learning rate in the fine-tuning phase is always $10^{-3}$ and is always fixed. Since we only fine-tune the model for only 1 or 5 epochs, we do not use a validation set for model selection.

## D ADDITIONAL EXPERIMENTS AND DISCUSSION

### D.1 PROPERTIES OF THE DIFFICULTY METRIC

To study the factors affecting the difficulty function defined in (1), let us denote by $d_1$, $d_2$ the difficulty functions obtained under two different training settings, such as different network architectures or adversarial attack strategies. We then define the difficulty distance (*D-distance*) between two such functions in the following equation. In this regard, the expected D-distance between two random difficulty functions is $0.375$.

$$D(d_1, d_2) = \mathbb{E}_{\boldsymbol{x} \sim U(\mathcal{D})} |d_1(\boldsymbol{x}) - d_2(\boldsymbol{x})| \ . \tag{45}$$

We then study the properties of the difficulty metric of Equation (1) by performing experiments on the CIFAR10 and CIFAR10-C (Hendrycks & Dietterich, 2019) dataset, varying factors of interest. In particular, we first study the influence of the network by using either a RN18 model, trained for either 100 or 200 epochs (RN18-100 or RN18-200), or a WRN34 model trained for 200 epochs (WRN34). To generate adversarial attacks, we make use of PGD with an adversarial budget based on the $l_\infty$ norm with $\epsilon = 8/255$. This corresponds to the settings used in other works (Hendrycks & Dietterich, 2019; Madry et al., 2018). The other hyper-parameters follow the general settings in Appendix C In the left part of Table 4, we report the D-distance for all pairs of settings. Each result is averaged over 4 runs, and the variances are all below $0.012$. The D-distances in all scenarios are all very small and close to $0$, indicating the architecture and the training duration have little influence on instance difficulty based on our definition.

| $d_1 \backslash d_2$ | RN18-100 | RN18-200 | WRN34 | | $d_1 \backslash d_2$ | Clean | FGSM | PGD |
|---|---|---|---|---|---|---|---|---|
| RN18-100 | 0.0189 | 0.0232 | 0.0355 | | Clean | 0.0189 | 0.0607 | 0.1713 |
| RN18-200 | 0.0232 | 0.0159 | 0.0299 | | FGSM | 0.0607 | 0.0843 | 0.1677 |
| WRN34 | 0.0355 | 0.0299 | 0.0178 | | PGD | 0.1713 | 0.1677 | 0.0857 |

Table 4: D-distances between difficulty functions in different settings, including different model architectures and training duration (left table), and different types of perturbations (right table).

We then perform experiments by varying the attack strategy using a RN18 network. As shown by the D-distances reported in the right portion of Table 4, the discrepancy between values obtained with clean, FGSM-perturbed and PGD-perturbed inputs is much larger, thus indicating that our difficulty function correctly reflects the influence of an attack on an instance. In addition, Table 5 demonstrates the D-distance between the difficulty functions based on clean instances, FGSM-perturbed instance, PGD-perturbed instances and different common corruptions from CIFAR10-C (Hendrycks & Dietterich, 2019)[9]. Note that (Hendrycks & Dietterich, 2019) only provides corrupted instances on the test set, so we train models on the clean training set and test model on corrupted test set in these cases. We use RN18 architecture and train it for 100 epochs in all cases, results are reported on the test set. Compared with the results in the left half of Table 4, the D-distance is much larger here. This indicates the difficulty function depends on the perturbation type applied to the input, including the common corruptions.

The results in Table 4 and 5 demonstrate that our difficulty metric mainly depends on the data and on the perturbation type; not the model architecture or the training duration.

---

[8]Data available for download on https://github.com/yguooo/semisup-adv. MIT license. Free to use.

[9]Data available for download on https://github.com/hendrycks/robustness. Apache License 2.0. Free to use.

| $d_1 \backslash d_2$ | brightness | contrast | defocus | elastic | fog | gaussian_blur |
|---|---|---|---|---|---|---|
| Clean | 0.1279 | 0.3219 | 0.2646 | 0.2115 | 0.2324 | 0.3069 |
| FGSM | 0.1303 | 0.3128 | 0.2642 | 0.2098 | 0.2289 | 0.3064 |
| PGD | 0.1873 | 0.3082 | 0.2616 | 0.2319 | 0.2414 | 0.2959 |

| $d_1 \backslash d_2$ | glass_blur | jpeg | motion_blur | pixelate | gaussian_noise | impulse_noise |
|---|---|---|---|---|---|---|
| Clean | 0.2809 | 0.1838 | 0.2520 | 0.2365 | 0.2999 | 0.2869 |
| FGSM | 0.2760 | 0.1853 | 0.2520 | 0.2417 | 0.2918 | 0.2807 |
| PGD | 0.2825 | 0.2026 | 0.2605 | 0.2551 | 0.2980 | 0.2866 |

| $d_1 \backslash d_2$ | saturate | shot_noise | snow | spatter | zoom_blur | speckle_noise |
|---|---|---|---|---|---|---|
| Clean | 0.1335 | 0.2832 | 0.2033 | 0.1930 | 0.2654 | 0.2829 |
| FGSM | 0.1329 | 0.2754 | 0.2003 | 0.1946 | 0.2657 | 0.2759 |
| PGD | 0.1932 | 0.2841 | 0.2148 | 0.2297 | 0.2711 | 0.2901 |

Table 5: D-distances between difficulty functions of vanilla / FGSM / PGD training and training based on 18 different corruptions on CIFAR10-C. We run each experiment for 4 times and report the average value.

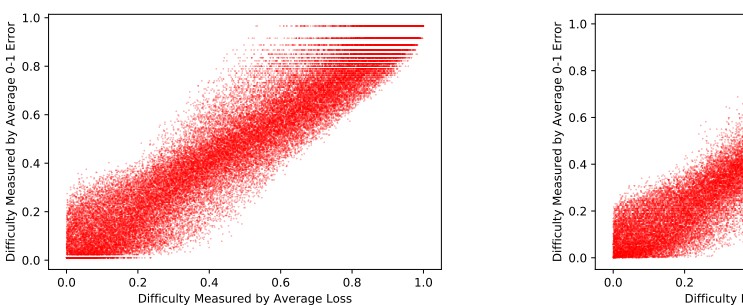

Figure 3: The relationship between the difficulty function based on the average loss values and the one based on the average 0-1 errors. The left figure is based on the RN18-200 model; the right figure is based on the WRN34 model. The correlation between these two metrics are 0.9466 (left) and 0.9545 (right), respectively.

In the definition of our difficulty metric in Equation (1), the difficulty of one instance is based on its average loss values during the training procedure. It is intuitive, because the values of the loss objective represents the cost that model needs to fit the corresponding data point. The bigger this cost is, the more difficulty this instance will be. To make the metric stable and prevent the metric from being sensitive to the stochasticity in the training dynamics, we use the average value of the loss objective for each instance to define its difficulty. In addition to the average loss objectives, we can also use the average 0-1 error to define the difficulty function. In Figure 3, we plot the relationship between the difficulty metric based on the average loss values and the one based on the average 0-1 error for instances in the CIFAR10 training set when we train a RN18-100 model and a WRN34 model. We can see a strong correlation between them for both models. The correlation of the difficulty measured by two metrics for the same instance is 0.9466 in the RN18-100 case and 0.9545 in the WRN34 case. The high correlation indicates we can use either metric to measure the difficulty. Since the loss objective values are continuous and finer-grained, we choose it as the basis of the difficulty function we use in this paper.

## D.2 EXAMPLES OF EASY AND HARD INSTANCES OF CIFAR10 AND SVHN

Figure 20 and 21 demonstrate the easiest and the hardest examples of each class from CIFAR10 and SVHN, respectively. The difficulty of these instances is calculated based on PGD attacks. We can see most easy examples are visually consistent, while most hard examples are ambiguous or even incorrectly labeled.

### D.3 TRAINING ON A SUBSET

**Different Optimization Method on Hard Instances** We find the failure of PGD adversarial training on the hardest 10000 instances of CIFAR10 training set does not arise from the optimizers. In Figure 5, we use SGD with different initial learning rates ("SGD, lr=1e-2" and "SGD, lr=1e-3") and the adaptive optimizer such as Adam (Kingma & Ba, 2014) ("Adam, lr=1e-4"). The learning rate of SGD optimizer decrease to its $1/10$ in the 100th and 150th epoch, while the learning rate of Adam optimizer is fixed during training. Although optimizers like Adam can make the model fit the training subset better, none of these methods can make the robust test accuracy significantly above the chance of random guesses, i.e., $10\%$.

**Longer Training Duration** In Figures 7 and 8, we conduct adversarial training for a longer duration until the loss on the hard training instances converges. Specifically, the model is trained for 600 epochs, with a learning rate is initialized to 0.1 and divided by 10 after every 100 epochs. In Figure 7, we adversarially train a RN18 model on the hardest 10000 training instances. Our conclusions from Section 4.1 still hold: Adversarial training on the hard instances leads to much more severe overfitting, greatly widening the generalization gap. In Figure 8, we adversarially train a RN18 model on the whole training set and calculate the average loss on the groups $\mathcal{G}_0$, $\mathcal{G}_3$, $\mathcal{G}_6$ and $\mathcal{G}_9$. Similarly to training for 200 epochs, the model first fits the easy training instances and then the hard ones. This can be seen by the fact that the average loss of $\mathcal{G}_9$ decreases much faster in the beginning and quickly saturates. In other words, the harder the group, the later we see a significant decrease in its average loss value. This observation is also consistent with our findings in Section 4.2.

**Results on SVHN dataset** Figure 6 demonstrates the learning curves of PGD adversarial training based on a subset of the easiest, the random and the hardest instances of SVHN dataset. We let the size of each subset be 20000, because the training set of SVHN is larger than that of CIFAR10. The model architecture is RN18 in these cases. We have the same observations here: training on the hardest subset yields trivial performance, training on the random subset has significant generalization decay in the late phase of training while training on the easiest subset does not.

**Different Values of $\epsilon$ and $l_2$-based Adversarial Budget** Figure 4 and Figure 9 demonstrate the learning curves of RN18 models under different adversarial budgets on CIFAR10, in both $l_\infty$ and $l_2$ cases. In $l_\infty$ cases, the adversarial budgets are $2/255$, $4/255$ and $6/255$; in $l_2$ cases, the adversarial budgets are 0.5, 0.75 and 1. With the increase in the size of the adversarial budget, we can see a clear transition from the vanilla training: more and more severe generalization decay when training on the random or the hardest subset.

**Training with Different Amount of Data** In Figure 10, we compare the learning curves of PGD adversarial training on increasing more training data, with the easiest ones coming first. If we do model selection on a validation set as in (Rice et al., 2020), the selected models are still better on both CIFAR10 and SVHN when they are trained with more data, although the final models in these cases are not necessarily better. The results indicate the hard instances are still useful to improve the model's performance, but we need to utilize them in a different way.

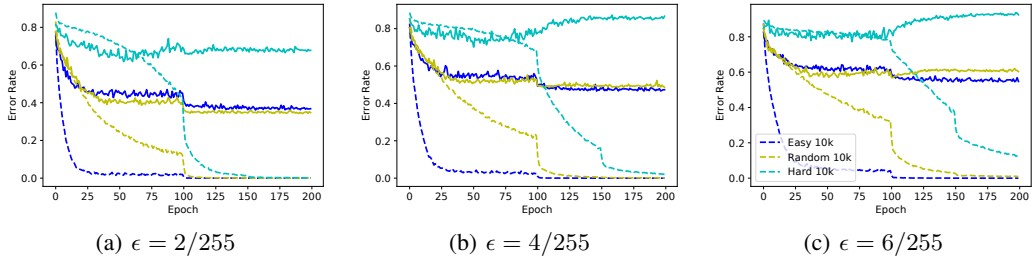

(a) $\epsilon = 2/255$      (b) $\epsilon = 4/255$      (c) $\epsilon = 6/255$

Figure 4: Learning curves of training on PGD-perturbed inputs against different sizes of $l_\infty$ norm based adversarial budgets using the easiest, the random and the hardest 10000 training instances. The instance difficulty is determined by the corresponding adversarial budget and is thus different under different adversarial budgets. The dashed lines are robust training error on the selected training set, the solid lines are robust test error on the entire test set.

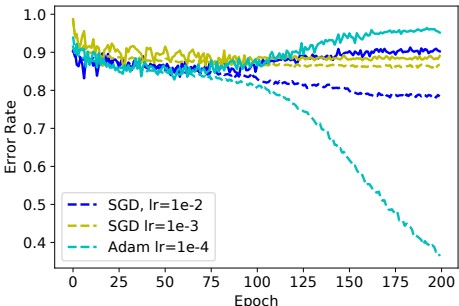

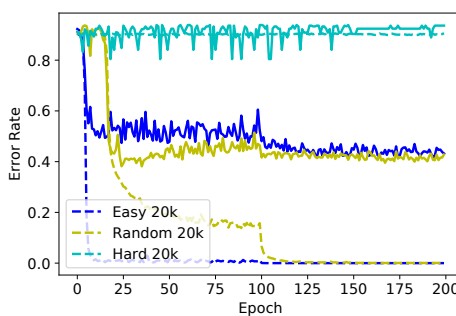

Figure 5: Learning curves of PGD adversarially trained models on the hardest 10000 instances in the CIFAR10 training set by different optimizers. The dashed lines are robust training error on the selected training instances, the solid lines are robust test error on the entire test set.

Figure 6: Learning curves obtained by training using the easiest, the random and the hardest 20000 instances of the SVHN training set. The training error (dashed lines) is the robust error on the selected instances, and the robust test error (solid lines) is always the error on the entire test set.

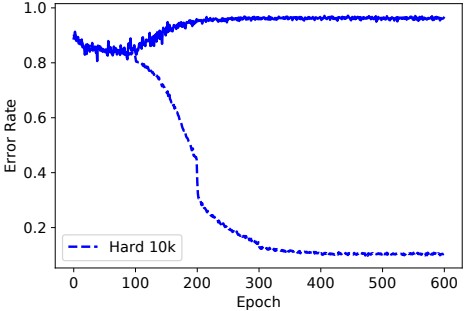

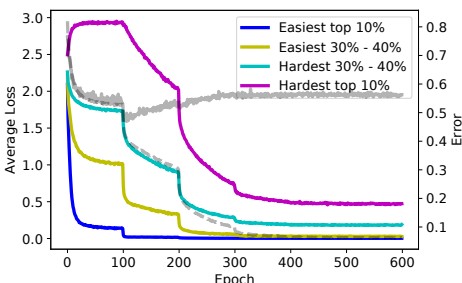

Figure 7: Learning curves of PGD adversarial training on the hardest 10000 training instances. The model is trained for 600 epochs. The training error (dashed lines) is the robust error on the selected instances, and the robust test error (solid lines) is always the error on the entire test set.

Figure 8: The left vertical axis represents the average loss of the training instances in the groups $\mathcal{G}_0$, $\mathcal{G}_3$, $\mathcal{G}_6$ and $\mathcal{G}_9$. The right vertical axis represents the robust error for the whole training (dashed grey line) and test (solid grey line) set. The model is trained for 600 epochs.

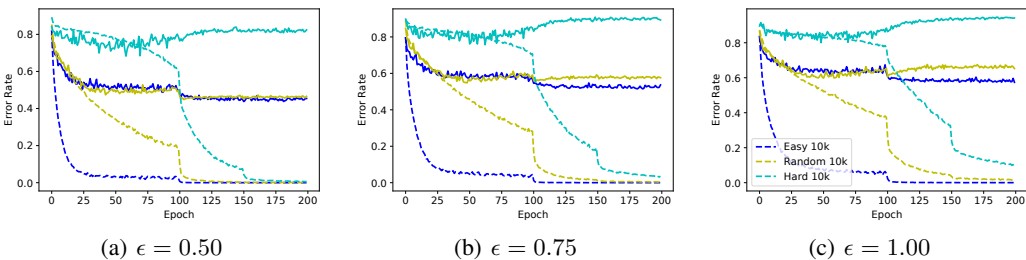

(a) $\epsilon = 0.50$           (b) $\epsilon = 0.75$           (c) $\epsilon = 1.00$

Figure 9: Learning curves of training on PGD-perturbed inputs against different size of $l_2$ norm based adversarial budgets using the easiest, the random and the hardest 10000 training instances. The instance difficulty is determined by the corresponding adversarial budget and is thus different under different adversarial budgets. The dashed lines are robust training error on the selected training set, the solid lines are robust test error on the entire test set.

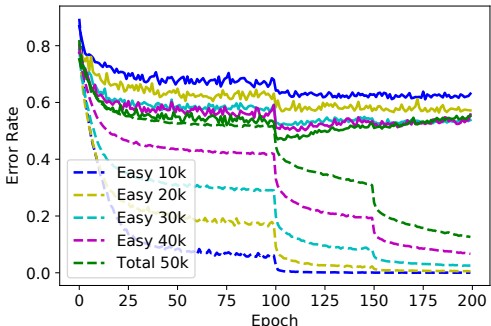 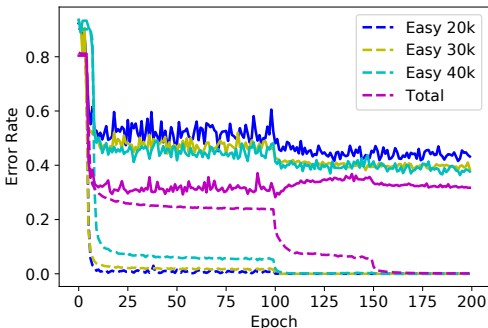

Figure 10: Learning curves of PGD adversarial training using increasing more training data in CIFAR10 and SVHN. The dashed lines represent the robust training error on the selected training instances; the solid lines represent the robust test error on the entire test set. Left: we use the easiest 10000, 20000, 30000, 40000 and the whole training set of CIFAR10. Right: we use the easiest 20000, 30000, 40000 and the whole training set of SVHN.

### D.4 LIPSCHITZ ANALYSIS

| Training Set | Lipschitz in $l_\infty$ Cases ($\times 10^4$) | | | Lipschitz in $l_2$ Cases ($\times 10^4$) | | |
|---|---|---|---|---|---|---|
| | $\epsilon = 2/255$ | $\epsilon = 4/255$ | $\epsilon = 8/255$ | $\epsilon = 0.50$ | $\epsilon = 0.75$ | $\epsilon = 1.00$ |
| Easy10K | 5.91 | 6.06 | 14.54 | 3.34 | 3.67 | 2.91 |
| Random10K | 28.98 | 79.96 | 93.63 | 30.01 | 31.28 | 39.34 |
| Hard10K | 72.42 | 117.60 | 567.24 | 60.62 | 80.06 | 77.55 |

Table 6: Upper bound of the Lipschitz constant under different settings of $\epsilon$ and training instances.

We conduct numerical analysis to validate Theorem 2 of Section 5.2. Calculating the exact Lipschitz constant of a neural network models is NP-hard, so we utilize the method introduced in (Scaman & Virmaux, 2018) to derive the upper bound of Lipschitz constants instead. We use CIFAR10 as the dataset and RN18 as the network architecture.

Table 6 demonstrates the upper bound of the Lipschitz constant of models trained in different settings. In the $l_\infty$ cases, we set $\epsilon$ to be $2/255$, $4/255$ and $8/255$; in the $l_2$ cases, we set $\epsilon$ to be $0.5$, $0.75$ and $1$. Due to the stochasticity introduced by power method, we run the algorithm in (Scaman & Virmaux, 2018) for 20 times and report the average, we find the variance is negligible. Based on the results in Table 6, it is clear that models adversarially trained by the hard training instances have much larger Lipschitz constant than the ones by the easy training instances in all cases.

Figure 11 demonstrates the curves of the Lipschitz upper bound when the model is adversarially trained by the easiest, the random and the hardest 10000 instances. The adversarial budget is $l_\infty$ norm based and $\epsilon = 8/255$. We can clearly see that as the training goes, the Lipschitz upper bound increases in all cases, which is consistent with our analysis in Section 5.2. In addition, compared with training on easy instances, the Lipschitz upper bound of the models adversarially trained on hard instances increases much faster.

### D.5 REVISITING EXISTING METHODS MITIGATING ADVERSARIAL OVERFITTING

Existing methods mitigating adversarial overfitting can be generally divided into two categories: one is to use adaptive inputs, such as (Balaji et al., 2019); the other is to use adaptive targets, such as (Chen et al., 2021b; Huang et al., 2020). Both categories aim to prevent the model from fitting hard input-target pairs. In this section, we pick one example from each category for investigation. We provide the learning curves of the methods we study in Figure 12. We use the same hyper-parameters as in these methods' original paper, except for the training duration and learning rate scheduler, which follow our settings. These methods clearly mitigate adversarial overfitting: The robust test error does not increase much in the late phase of training, and the generalization gap is much smaller that that of PGD adversarial training.

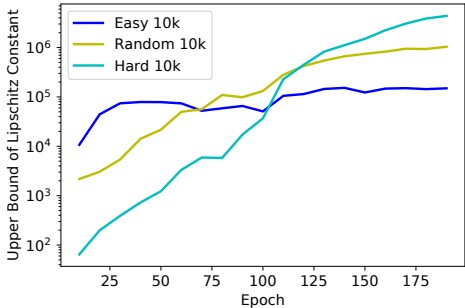

Figure 11: The curves of the Lipschitz upper bound when the model is adversarially trained by the easiest, the random and the hardest 10000 instances. The y-axis is log-scale.

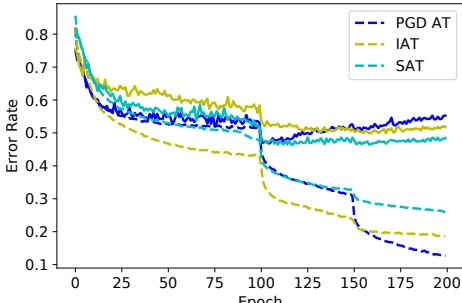

Figure 12: Learning curves of PGD adversarial training (PGD AT), instance-adaptive training (IAT) and self-adaptive training (SAT). Dashed lines and solid lines represent the robust training error and the robust test error, respectively.

**Instance-Adaptive Training** Using an instance-adaptive adversarial budget has been shown to mitigate adversarial overfitting and yield a better trade-off between the clean and robust accuracy (Balaji et al., 2019). In instance-adaptive adversarial training (IAT), each training instance $\boldsymbol{x}_i$ maintains its own adversarial budget's size $\epsilon_i$ during training. In each epoch, $\epsilon_i$ increases to $\epsilon_i + \epsilon_\Delta$ if the instance is robust under this enlarged adversarial budget. By contrast, $\epsilon_i$ decreases to $\epsilon_i - \epsilon_\Delta$ if the instance is not robust under the original adversarial budget. Here, $\epsilon_\Delta$ is the step size of the adjustment.

We use the same settings as in (Balaji et al., 2019) except that we use the same number of training epochs and learning rate scheduling as the one in other experiments for fair comparison. Specially, we set the value of $\epsilon$ and $\epsilon_\Delta$ to be $8/255$ and $1.9/255$, respectively, same as in (Balaji et al., 2019). The first 5 epochs are warmup, when we use vanilla adversarial training (Madry et al., 2018).

Figure 13 demonstrate the relationship between the instancewise adversarial budget $\epsilon_i$ and the corresponding instance's difficulty $d(\boldsymbol{x}_i)$. It is obvious that they are highly correlated: the correlation is $0.844$. Therefore, instance-adaptive training adaptively uses smaller adversarial budgets for hard training instances, which prevents the model from fitting hard input-target pairs.

**Self-Adaptive Training** Self-adaptive training (SAT) (Huang et al., 2020) solves the adversarial overfitting issue by adapting the target. By contrast with common practice consisting of using a fixed target, usually the ground-truth, SAT adapts the target of each instance to the model's output. Specifically, after a warm-up period, the target $\boldsymbol{t}_i$ for an instance $\boldsymbol{x}_i$ is initialized as a one-hot vector by its ground-truth label $y_i$ and updated in an iterative manner after each epoch as $\boldsymbol{t}_i \leftarrow \rho\boldsymbol{t}_i + (1-\rho)\boldsymbol{o}_i$. Here, $\rho$ is a predefined momentum factor and $\boldsymbol{o}_i$ is the output probability of the current model on the corresponding clean instance. SAT uses the loss of TRADES (Zhang et al., 2019b) but replaces the ground-truth label $y$ with the adaptive target $\boldsymbol{t}_i$: $\mathcal{L}_{SAT}(\boldsymbol{x}_i) = \mathcal{L}(\boldsymbol{x}_i, \boldsymbol{t}_i) + \lambda \max_{\Delta_i \in \mathcal{S}(\epsilon)} KL(\boldsymbol{o}_i \| \boldsymbol{o}_i')$, where $KL$ refers to the Kullback–Leibler divergence and $\lambda$ is the weight for the regularizer. Furthermore, SAT uses a weighted average to calculate the loss of a mini-batch; the weight assigned to each instance $\boldsymbol{x}_i$ is proportional to the maximum element of its target $\boldsymbol{t}_i$ but

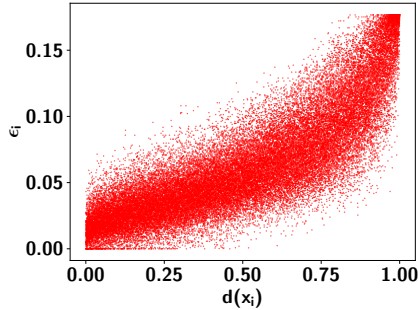

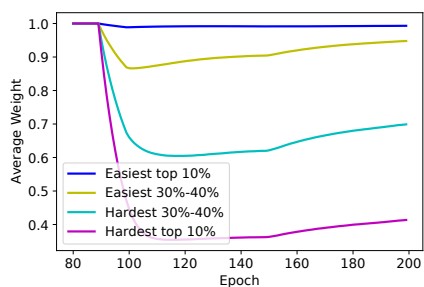

Figure 13: The instance adaptive adversarial budget size $\epsilon_i$ for the training set of CIFAR10 as a function of the instance difficulty $d(\boldsymbol{x}_i)$. The model architecture is RN18 and the value of $\epsilon$ is $8/255$.

Figure 14: Average weights of different groups in the training set of CIFAR10 during training. During the warmup period (first 90 epochs), the weight for every training instance is 1. The model architecture is RN18.

normalized to ensure that all instances' weights sum up to 1. By weighted averaging, the instances with confident predictions are strengthened, whereas the ambiguous instances are downplayed.

Similarly, we use the same settings as in (Huang et al., 2020) except we use the same number of training epochs and learning rate scheduling: we train the model for 200 epochs and the first 90 epochs are the warmup period.

Figure 14 demonstrates the average weight assigned to instances belonging to the group $\mathcal{G}_0, \mathcal{G}_3, \mathcal{G}_6$ and $\mathcal{G}_9$. It is clear that the hard instances are assigned much lower weights than the easy instances. For example, the weight assigned to $\mathcal{G}_0$, the easiest $10\%$ training instances, is close to 1, while the weight assigned to $\mathcal{G}_9$, the hardest $10\%$ training instances, is only around $0.4$.

Furthermore, Figure 15 shows the accuracy of the group $\mathcal{G}_0, \mathcal{G}_3, \mathcal{G}_6$ and $\mathcal{G}_9$ during training using the original ground-truth label $\mathbf{1}_y$ and the adaptive target $\boldsymbol{t}$, respectively. For the adaptive target, one adversarial instance $\boldsymbol{x}'$ is considered to be correctly classified if and only if $\arg\max_i f_{\boldsymbol{w}}(\boldsymbol{x}')_i = \arg\max_i \boldsymbol{t}_i$. For easy instances, $\mathbf{1}_y$ is mostly close to $\boldsymbol{t}$, so accuracy in both cases is high and the gap between them is small. For hard instances, $\mathbf{1}_y$ is usually not consistent with $\boldsymbol{t}$, while accuracy under the adaptive target $\boldsymbol{t}$ is much higher than the group-truth label $y$. This indicates self-adaptive training makes adaptive target easier to fit for the originally hard instances.

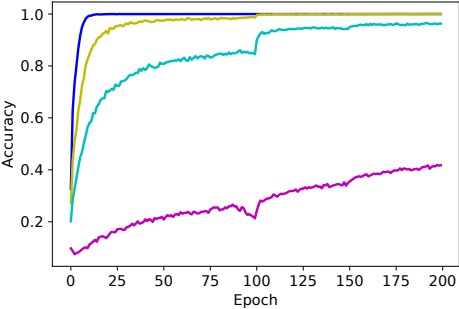

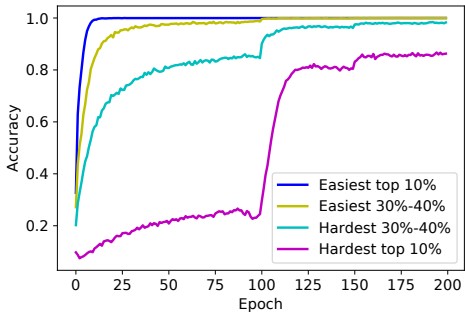

Figure 15: Robust training accuracy during training when we use the original groundtruth label (left) or use the adaptive target calculated during training (right).

## D.6 EXTRA RESULTS AND DISCUSSION ON FAST ADVERSARIAL TRAINING

We also conduct ablation study in the context of fast adversarial training. In Figure 16, we change the value of $\beta$ in our algorithm (pseudocode in Algorithm 1) and plot the learning curves. Lower the value of $\beta$ is, more weights assigned to the adaptive part of the target: $\beta = 0$ means we directly utilize the moving average target as the final target, $\beta = 1$ means we use the one-hot groundtruth

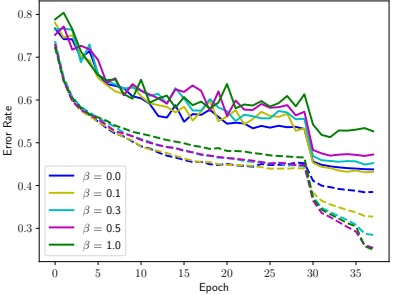

Figure 16: The learning curves with different values of $\beta$. The solid curve and the dashed curve represent the robust test error and the robust training error, respectively.

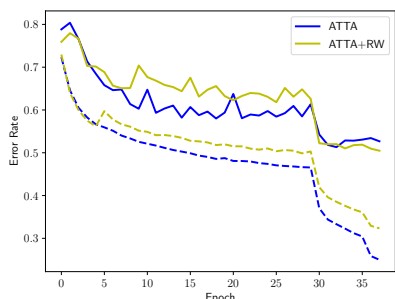

Figure 17: The learning curves of ATTA with and without reweighting. The solid curve and the dashed curve represent the robust test error and the robust training error, respectively.

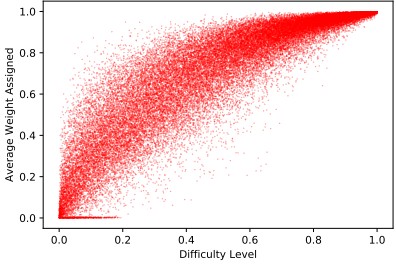

Figure 18: The relationship between the difficulty value and the weight assigned to each instances when using reweighting. We use the average weight across epochs. The correlation between them is 0.8900.

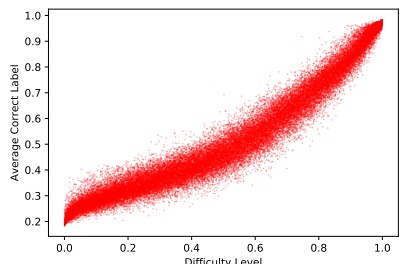

Figure 19: The relationship between the difficulty value and the average value of the true label's probability when using the adaptive targets. The correlation between them is 0.9604.

label. Figure 16 clearly shows us the generalization gap decreases with the decrease in $\beta$. That is to say, the adaptive target can indeed improve the generalization performance.

Figure 17 compare the learning curves of ATTA (Zheng et al., 2020) with and without reweighting. The first 5 epochs are the warmup period. The results confirm that the reweighting scheme can prevent adversarial overfitting and decrease the generalization gap.

To confirm that the algorithm we use is consistent with our theoretical and empirical analysis in Section 4 and 5, we study the relationship between the instance difficulty and the weight assigned to them when using reweighting, as well as the soft target when using adaptive targets. Since the evaluation of model robustness is based on the PGD attack, the difficulty value here is based on the PGD perturbation. In Figure 18, we demonstrate the relationship between the difficulty value and the average assigned weight for each instance when using reweighting. We calculate the correlation between these two values on the training set, it is 0.8900. This indicates we indeed assign smaller weights for hard training instances and assign bigger weights for easy training instances. In Figure 19, we show the relationship between the difficulty value and the average value of the true label's probability in the soft target when we use the adaptive targets. Similarly, we calculate the correlation between these two values on the training set, it is 0.9604. This indicates the adaptive target is similar to the ground-truth one-hot target for the easy training instances, while the adaptive target is very different from the ground-truth one-hot target for the hard training instances. This means, adaptive targets prevent the model from fitting hard training instances while encourage the model to fit the easy training instances.

## D.7 EXTRA RESULTS AND DISCUSSION ON ADVERSARIAL FINETUNING

We conduct ablation study and the results are demonstrated in Table 7. It is clear that both reweighting and the KL regularization term benefit the performance of the finetuned model.

| Duration | Method | AutoAttack | Duration | Method | AutoAttack |
|---|---|---|---|---|---|
| **WRN34 on CIFAR10, $\epsilon = 8/255$** | | | **RN18 on SVHN, $\epsilon = 0.02$** | | |
| No Fine Tuning | | 52.01 | No Fine Tuning | | 67.77 |
| 1 Epoch | Vanilla AT | 54.11 | 1 Epoch | Vanilla AT | 70.81 |
| | RW | 54.69 | | RW | 70.83 |
| | KL | 54.73 | | KL | 72.29 |
| | RW + KL | 54.69 | | RW + KL | 72.53 |
| 5 Epoch | Vanilla AT | 55.49 | 5 Epoch | Vanilla AT | 72.18 |
| | RW | 56.41 | | RW | 72.72 |
| | KL | 56.55 | | KL | 73.17 |
| | RW + KL | 56.99 | | RW + KL | 73.35 |

Table 7: Ablation study on the influence of reweighting (RW) and the KL regularization term (KL) in the performance of adversarial finetuning with additional data.

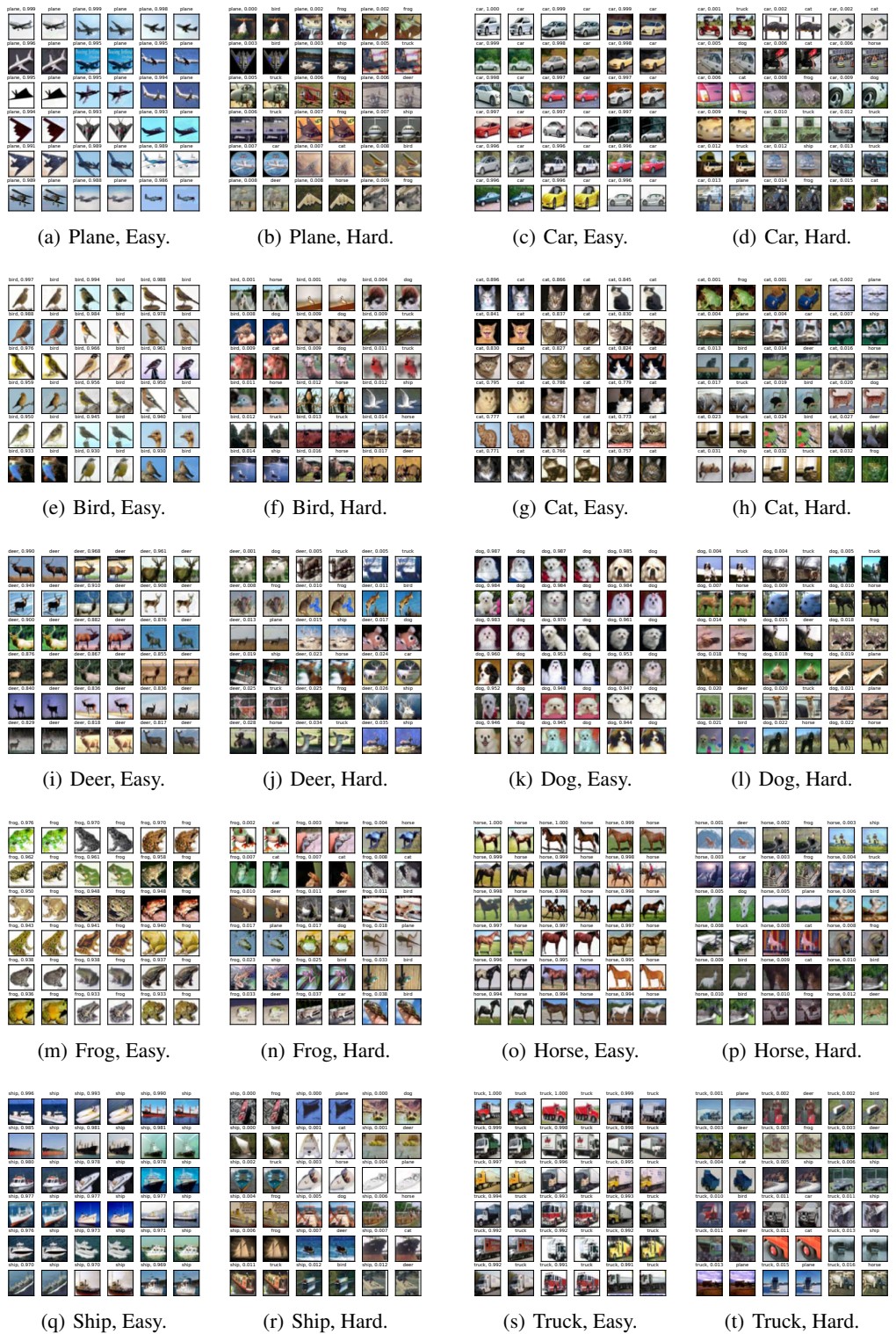

Figure 20: Easy and hard examples in each category of CIFAR10 dataset. In each subfigure, odd columns present the original images, and even columns present the PGD-perturbed images. Above each image, we provide the normalized difficulty defined in Equation (1) as well as the labels: true labels for the original images and the predicted labels for the perturbed images.

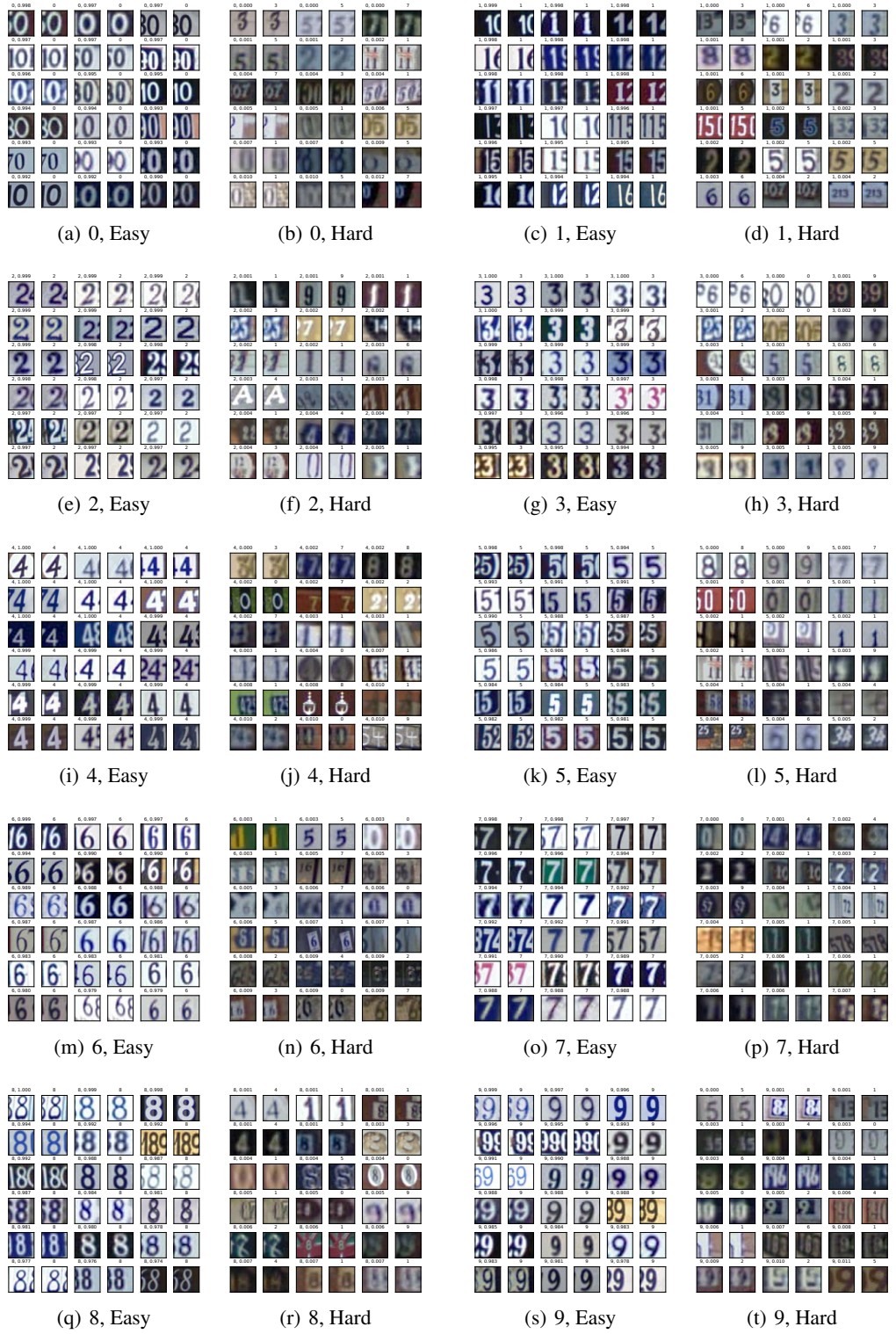

Figure 21: Easy and hard examples in each category of SVHN dataset. In each subfigure, odd columns present the original images, and even columns present the PGD-perturbed images. Above each image, we provide the normalized difficulty defined in Equation (1) as well as the labels: true labels for the original images and the predicted labels for the perturbed images.

