# OpenReview forum: "On the Impact of Hard Adversarial Instances on Overfitting in Adversarial Training"
_ICLR.cc/2022/Conference — ICLR 2022 Submitted_

### Official Review · Reviewer_crXx · 2021-10-29

**Correctness:** 3
**Technical Novelty And Significance:** 2
**Empirical Novelty And Significance:** 2
**Recommendation:** 3
**Confidence:** 4

**Main Review:**

Strengths:

1.	This paper shows that hard adversarial instances lead to degraded generalization performance in adversarial training based on the measurement of instance difficulty.

2.	This paper provides extensive theoretical analysis that shows fitting hard adversarial data hurts adversarial robustness.

Questions:

1.	I am concerned about the meaning of 'model-agnostic' which is one of the property of the metric for instance difficulty.  The calculation of loss for each instance depends on the parameters and the optimization for each model. Therefore, how can you ensure for each model and each setting of training, the difficulty of a particular instance keep the same?

2.	In Sec.4.2, this paper claims that adversarial overfitting arises from the model’s attempt to fit the hard adversarial instances. However, it is confusing that the easy instance (yellow lines in the right panel of Figure 2) has the similar behavior with the hard instance (red lines in the right panel of Figure 2) and the model fits the easy instances (yellow lines in the left panel of Figure 2) and hard instances (red lines in the left panel of Figure 2) almost simultaneously. Therefore, it seems that adversarial overfitting happens when the model’s attempt to fit the easy and hard adversarial instances.

3.	Theorem 2 and Theorem 3 both indicate that with the increase of instance difficulty, the adversarial robustness becomes lower. However, it is not clear how can those theorems show the relationship between robust overfitting and instance difficulty.

4.	Although the effectiveness of two tricks proposed in Sec.6 are empirically validated, the motivation for these two tricks is not clear. The reweight strategy gives lower weight for the instance whose predicted probability is lower. However, it is hard to say the instance whose predicted probability is low is the hard instance measured by the proposed metric. Therefore, the proposed method seems to be hardly related to the metric.


**Summary Of The Paper:**

This paper proposes a metric for measuring the difficulty of instances. The authors empirically show that hard instances lead to the issue of robust overfitting. Theoretical analysis on logistic regression and general nonlinear model indicates that as the increase of instance difficulty, the adversarial vulnerability becomes higher. Finally, this paper proposes a reweight strategy and the trick of adaptive label for improving the adversarial robustness.

**Summary Of The Review:**

Overall, this paper investigates the impact of hard instance on adversarial training. However, I am not convinced of the novelty and the correctness of the proposed method.

---

> ### Author Response · Authors · 2021-11-20
> **Response to Reviewer crXx (1 / 2)**
>
> We thank the reviewer for the constructive comments. The changes in the manuscript are highlighted in blue in the revision. Below are our point-to-point responses to the reviewer
>
> 1. **Concerns about “model-agnostic” claim**
>
> We thank the reviewer for pointing this out and admit our original statement was not very rigorous. We have revised the corresponding claims. The metric in definition (1) indeed depends on the model and its training method in theory. However, **as shown in Appendix D.1**, with a fixed adversarial budget and perturbation type, the model architecture and the training duration have very little impact on the difficulty function defined in Equation (1). **As a result, the difficulty function can be considered approximately “model-agnostic” and a property of the data distribution under adversarial attack.**
>
>
> 2. **Concerns about Figure 2**
>
> For Figure 2(a), we agree that the loss values for most groups are decreasing during training, but **the decreasing speed (the slope of the curve) is different for different groups in different training phases**. In the early phase of training, the loss of easy instances (blue, yellow lines) decreases faster, and the loss of hard instances decreases more slowly (light blue) or even increases (purple). By contrast, in the late training phase, the loss of the hard instances decreases faster than the easy ones (decrease speed: purple > light blue > yellow > blue). This indicates that the model focuses on fitting the easy training instances first and then the hard training instances. In Figure 8 of Appendix D.3 in our revised manuscript, we train the model for a longer time (600 epoch), and observe that **the harder the group is, the later we see a significant decrease in its average loss values.**
>
> For Figure 2(b), all groups of training instances eventually have similar feature magnitudes, but in the early phase, the magnitudes of the extracted features for the easiest training instances is much larger than the others. As training proceeds, the model starts to fit increasingly difficult training instances. In Figure 2(b), the blue line (the easiest top 10%) has the largest slope first, then the yellow line (the easiest 30% - 40%) has the largest slope around the 100th to 150th epoch, and finally the light blue line (the hardest 30% - 40%) has the largest slope after 150th epoch. This observation is consistent with Figure 2(a): **the harder the group is, the later it has the largest increasing slope in the feature magnitude**. All these observations are consistent with our claim: the model first fits the easiest training instances and their features, and then fits harder and harder training instances. Eventually, the model indeed fits both easy and hard training instances, but since the model fits the easy instances very quickly, adversarial overfitting only happens when the model tries to fit the hard instances.

---

> ### Author Response · Authors · 2021-11-20
> **Response to Reviewer crXx (2 / 2)**
>
> 3. **Clarification of Theorem 2, Theorem 3 and how they are connected with adversarial overfitting**
>
> Our theoretical analysis in both the linear and nonlinear cases studies how the difficulty of training instances affects the generalization performance on the whole data distribution, which contains both easy and hard examples. This is consistent with the settings of our empirical study in Section 4.1, especially what we show in Figure 1. **In both our theoretical and empirical analyses, we adversarially train a model using a subset of the training data, and then evaluate the model’s performance on the whole data distribution. All the results show that training on hard adversarial instances leads to a more severe adversarial overfitting and a larger generalization gap**. In our theoretical analysis, the difficulty level of the training instances is depicted as the norm of the centroid $r_l$ in the linear case and the conditional variance $\sigma_l$ in the nonlinear one. An increase in the norm of $r_l$ and a decrease in the conditional variance $\sigma_l$ can lead to a decrease in the loss values. Therefore, they are valid metrics to measure difficulty.
>
> In the linear case, we calculate the exact analytical form of the robust test error on the Gaussian mixture distribution assumed in Equation (3). Since we assume that the training instances are linearly separable in the over-parameterized case, the robust test error is then equivalent to the generalization gap. The analytical form of the robust test error, shown in Equation (4) in Theorem 2, indicates that the robust test error, i.e. the generalization gap, increases with a decrease in $r_l$, in turn showing that harder training instances lead to larger generalization gaps. In Corollary 1, we study the effect of the adversarial budget size on the sensitivity of the generalization gap to the difficulty of the training instances. Our results show that the difference in generalization gap between models trained by easy and hard instances is larger when the adversarial budget size $\epsilon$ is larger. This is consistent with our empirical findings in Figure 4 and Figure 9 in Appendix D.3 of the revised manuscript.
>
> In the nonlinear case, we use the Lipschitz constant as a proxy to measure the robust test error. Theorem 3 indicates that when the mean squared error on the adversarial training set is $C$, then with high probability the Lipschitz constant is upper bounded by Inequality (7). We note that the upper bound, i.e., the right hand side of (7), increases with $\gamma$, and therefore increases with $\sigma_l$, $\epsilon$ and decreases when $C$ increases. **As a result, as adversarial training processes, the adversarial loss $C$ on the training set becomes smaller and then the Lipschitz constant upper bound increases, which leads to a larger robust test error and a larger generalization gap. Furthermore, the Lipschitz constant upper bound also increases with the adversarial budget size $\epsilon$ and the training instances’ difficulty measured by $\sigma_l$, indicating that adversarial overfitting is more severe under a large adversarial budget and harder adversarial training instances.** This is exactly the observation of Figure 1, Figure 4 and Figure 9. The conclusion of Theorem 3 is valid on general nonlinear models and adversarial budgets based on any $l_p$ norm, so **our theoretical results are general applicable**. The robust test error, which also indicates the generalization gap since the training error is small (assumed by small $C$ in our Theorem), is depicted by an upper bound of the model’s Lipschitz constant. The numerical analysis in Appendix D.4 and especially the results in Table 6 and Figure 11 confirms the validity of Theorem 3 empirically.
>
> 4. **Relationship between Section 6 and the metric proposed**
>
> The lower predicted probability of the correct label indicates the larger cross-entropy loss for the instances. Since our metric introduced in Section 3 is based on the instance-wise loss, it is natural to consider those with lower predicted probability of the correct label as hard training instances. Lower weight given to hard instances means we prevent the model from fitting hard adversarial instances. **To further validate this, we add Figure 18 and 19 in our revised manuscript.** In Figure 18, we show the relationship between the difficulty of the training instance and the weight assigned to them when using reweighting. We can see the strong correlation between them: the correlation is $0.8900$. In Figure 19, we calculate the average probability of the true label in the soft target when we use the adaptive targets. We again show it is highly correlated with the instance difficulty: the correlation is $0.9604$. Figure 18 and 19 show both reweighting and adaptive targets can prevent from fitting adversarial hard instances. In addition, in Appendix D.5, we show our findings also holds for existing methods in standard adversarial training.

---

### Official Review · Reviewer_oU1N · 2021-11-02

**Correctness:** 3
**Technical Novelty And Significance:** 2
**Empirical Novelty And Significance:** 1
**Recommendation:** 5
**Confidence:** 5

**Details Of Ethics Concerns:**

No Ethics Concerns.

**Main Review:**

Strengths:

The authors first gave experiments and theoretical analysis to illustrate the relationship between hard samples and robust over-fitting, which is an interesting observation. Then, the proposed algorithm is reasonable and makes sense.

Weaknesses:

1. My primary concern is about the novelty. The difficulty level of samples has been considered in the adversarial training, especially the strategy that considers the weight of examples (like MART and SAT). I understand that the author constructs a new adversarial training by observing the relationship between hard samples and robust over-fitting. However, the proposed algorithm is very similar to self-adaptive training (SAT). Even in the experiments, the author did not compare and analyze the SAT algorithm.

2. It is not clear how to distinguish hard adversarial samples. The authors first said they provide a model-agnostic metric, which can measure the difﬁculty of an instance. But, it is hard to understand how to use this metric to guide adversarial training. In my opinion, section 6 does not seem to have much to do with the previous observation.

3. I carefully compare and analyze the difference between the algorithm in this paper and SAT, and I think it is mainly on the final adaptive target. So, if we also add this line to SAT, what will be the result?

**Summary Of The Paper:**

This paper began by analyzing the influence of hard adversarial examples. And, they found that hard adversarial examples may be the major reason leading to robust overfitting. Then, based on these observations and analysis, the authors further introduced a Fast Adversarial Training scheme, which can achieve competitive results compared to baselines.

**Summary Of The Review:**

Through experiments and theoretical analysis, this paper concludes that hard adversarial samples will lead to overfitting. However, the introduced algorithm is not novel enough, which is very similar to the existing self-adaptive training. Even in the experiments, the corresponding experiments are also missing. Based on these, I recommend ''marginally below the acceptance threshold''.

---

> ### Author Response · Authors · 2021-11-19
> **Response to Reviewer oU1N (1 / 2)**
>
> We thank the reviewer for the constructive comments, which helped us to polish the paper. The changes in the manuscript are highlighted in blue. Below are the point-to-point responses.
>
> 1. **Novelty and contributions of this paper**
>
> First, we need to clarify that the main focus of this paper is to point out that fitting hard training instances leads to overfitting and that this concern is particularly severe in adversarial training (Section 4 and Section 5). The main goal of Section 6 is not to design a new algorithm to achieve state-of-the-art performance, but to show that 1) existing methods claiming to mitigate adversarial overfitting all implicitly prevent the model from fitting hard adversarial instances; 2) these ideas are not only applicable to standard adversarial training, but also more broadly to, e.g., fast adversarial training and adversarial finetuning; the performance gain is even larger in these cases.
>
> **The main contribution of this paper is to rigorously and extensively demonstrate that adversarial overfitting arises from fitting adversarial examples. Our claim is stronger than the previous MART paper, which encourages to treat misclassified instances differently and has no theoretical analysis. By contrast, we validate our claims both empirically and theoretically, both in the $l_\infty$ and general $l_p$ norm cases, and for both linear and general nonlinear models.** Specifically, our theoretical analysis (Section 5) rigorously explains the empirical observations of Section 4. In the linear case (Section 5.1), we calculate the exact analytical form of the generalization gap. We show that the difference in the generalization gap between models trained using easy and hard instances becomes larger with the size of the adversarial budget (Corollary 1), indicating that adversarial overfitting is more severe when adversarially training the model against larger perturbations. Section 5.2 uses the Lipschitz constant as a proxy to depict the robust test accuracy of the network. Theorem 3 explains the observation of “degraded adversarial accuracy in the later phase of training” from a theoretical standpoint. We show that, when the adversarial training loss is smaller (smaller C in Theorem 3), the Lipschitz constant and thus the test robust accuracy becomes larger, which means degraded performance on the test set and a larger generalization gap. Furthermore, we show that this gap is larger when using hard training adversarial examples (i.e., large $\sigma_l$) and a large adversarial budget (i.e., large $\epsilon$) by Theorem 3 and its analysis in Section 5.2. This is consistent with our empirical observations in Section 4. **Therefore, the focus of this work is not to propose new algorithms to mitigate adversarial overfitting, but to investigate the fundamental reason behind this phenomenon and prove it theoretically and empirically.**
>
> 2. **Connections between the difficulty metric and the method in Section 6**
>
> First, we analyze the existing approaches to mitigate adversarial overfitting in Appendix D.5 and find they all implicitly prevent from fitting hard adversarial training instances (Figure 13, 14 and 15 in the *revised manuscript*). **Furthermore, to better show that the method we use in Section 6 is consistent with our findings in Section 4 and Section 5, we added Figure 18 and Figure 19 in Appendix D.6 in our revised manuscript.** In Figure 18, we show the relationship between the difficulty of the training instance and the weight assigned to it when using reweighting in our method. We can see a strong correlation of $0.8900$ between them. This indicates that our reweighting scheme assigns smaller weights for hard instances and thus avoids fitting them during training. In Figure 19, we calculate the average probability of the correct label in the soft target when using the adaptive targets. We again show a high correlation ($0.9604$) with the instance difficulty. This indicates that, when using adaptive targets, the easy instances are assigned a target similar to the ground-truth one-hot label, while the hard instances are not. Adaptive targets prevent the model from fitting hard adversarial input-target pairs.

---

> ### Author Response · Authors · 2021-11-19
> **Response to Reviewer oU1N (2 / 2)**
>
> 3. **Comparison between the SAT paper and the method used in this paper**
>
> Our idea of using adaptive targets in fast adversarial training is indeed based on SAT, and we clarified this in our revision. Algorithm 1 in Appendix C.2 nonetheless differs from SAT in three aspects:
> * We use the cross-entropy loss instead of the TRADES-based loss in SAT, because TRADES needs to calculate the output of the clean data, which increases the computational complexity.
>  *To also save computational complexity, the adaptive targets we use are based on the probability of the adversarial inputs, whereas SAT uses the probability of the clean inputs.
>  * We utilize ATTA as the initial point for the adversarial perturbations, whereas SAT does not.
>
> In Appendix D.5, we revisit the existing methods mitigating adversarial overfitting including SAT. In our revised version, we also provide the learning curve of SAT in Figure 12. We show that the methods using adaptive targets, such as SAT, can indeed mitigate adversarial overfitting. However, the performance improvement on the test set is marginal in standard adversarial training: When using TRADES as in SAT, the performance improvement is around 1% (Figure 4 in the SAT paper). By contrast, in fast adversarial training, the performance improvement is much larger; according to Table 1, ATTA with adaptive targets improves ATTA by 6.7%.

---

### Official Review · Reviewer_dYnb · 2021-11-02

**Correctness:** 3
**Technical Novelty And Significance:** 3
**Empirical Novelty And Significance:** 3
**Recommendation:** 6
**Confidence:** 4

**Main Review:**

Strength:
+ The topic studied in this paper is interesting and important. It can help practitioners better understand how training data influences the overfitting of adversarial training.
+ This paper provides a theoretical explanation for why hard examples hurt the generalization gap.
+ The evaluation results empirically show that adaptive training examples can improve robustness.

Weakness:
- The paper is a little bit overclaimed in the abstract. By definition, the difficulties of training examples are defined with losses of corresponding models, which means that it is not model agnostic. Same training samples could have different difficulties for different models. The method to calculate the metric is model agnostic, but the metric is not model agnostic.

- In Section 3, the authors define the difficulty metric for training samples. One of my recommendations is to explain more about how and why the metric is defined.

- Figures 1(a) and 2(a) show average losses of models trained on data with different difficulties, but I find that models with harder training data don't reach a plateau. A potential reason could be that models need more training epochs to converge on hard training examples. With enough training epochs, it is possible for models training on hard examples to get similar average losses.

- The proposed training method is only evaluated on fast adversarial training and adversarial fine-tuning with additional data, but not on standard adversarial training. From my understanding, we can also combine this method with standard adversarial training. If the evaluation results also show improvements on standard adversarial training, the paper will be more compelling.


**Summary Of The Paper:**

This paper proposes a metric to measure the difficulty of training examples and find that hard training examples influence the generalization of adversarial training and cause overfitting in adversarial training. The authors also provide a theoretical analysis.
To mitigate the issue caused by hard training examples, the authors propose to assign different weights for training examples based on loss values. The evaluation results show that the proposed training methods can improve robustness in fast adversarial training and adversarial fine-tuning with additional data.


**Summary Of The Review:**

This paper studies an interesting topic. It will be great that the method is also evaluated with other SOTA standard adversarial training methods.

---

> ### Author Response · Authors · 2021-11-19
> **Response to Reviewer dYnb**
>
> We thank the reviewer for the constructive comments. Our changes in the manuscript are highlighted in blue. Below are our responses to the questions raised by the reviewer.
>
> 1. **Overclaimed ”model-agnostic”**
>
> We thank the reviewer for pointing this out and admit our original statement was not very rigorous. We have revised the corresponding claims. The metric in definition (1) indeed depends on the model and its training method in theory. However, as shown in Appendix D.1, with a fixed adversarial budget and perturbation type, the model architecture and the training duration have very little impact on the difficulty function defined in Equation (1). **As a result, the difficulty function can be considered approximately “model-agnostic” and a property of the data distribution under the adversarial attack.**
>
> 2. **Motivation of the difficulty metric**
>
> Using the instance-wise loss values as the basis for the difficulty function is intuitive, because the loss objective by definition is the cost that the model needs to fit a data sample. The bigger this cost is, the more difficult this instance will be. To make the metric stable and prevent it from being overly sensitive to the stochasticity of the training dynamics, we use the average value of the loss objective during the training procedure to define the difficulty function. Note that we can also base our metric on the average 0-1 error. In Figure 3 of Appendix D.1 in our *revised manuscript*, we show a high correlation (~0.95) between the difficulty function based on the average 0-1 error and on the average loss value. Since the loss values are continuous and thus finer-grained, we chose them to define our difficulty metric in Equation (1).
>
> 3. **Training models in Figure 1(a) and Figure 2(a) until the loss reaches a plateau.**
>
> We agree with the reviewer that the loss values for hard adversarial instances do not converge in 200 epochs. Therefore, **we trained the model for 600 epochs in the same settings and provide the resulting learning curves in Figure 7 and Figure 8 of Appendix D.3 in the *revised manuscript*.** From Figure 7, we see that the robust test accuracy of the model trained on hard instances remains very low and decreases during training, indicating that training on hard instances leads to very severe overfitting. This observation is consistent with Section 4.1. In Figure 8, the average loss of $\gG_9$, the easiest group, decreases much faster in the beginning and quickly saturates. Furthermore, the harder the group, the later we see a significant decrease in its average loss value. Considering that robust overfitting happens and becomes more severe in the late training phase, this observation is consistent with our conclusions in Section 4.2: fitting hard adversarial instances leads to robust overfitting.
>
> 4. **Evaluations on standard adversarial training**
>
> We briefly revisit the existing methods mitigating adversarial overfitting in standard adversarial training in Appendix D.5 and show that they all implicitly prevent the model from fitting hard adversarial examples. If we adapt these methods to the cases of fast adversarial training and adversarial finetuning (such as ATTA with adaptive targets), we can see it not only avoids adversarial overfitting, but also improves the performance. **Note that in these two cases, the performance is considerably improved (ATTA with adaptive targets improves ATTA by 6.7% in Table 1), while the performance in the case of standard adversarial training is only marginally improved (SAT improves standard adversarial training by ~1% in the SAT paper).**
>
> **Contributions and focus of this paper**
>
> **The main focus of this paper is to demonstrate that hard training instances lead to overfitting, and that this effect is particularly prominent in adversarial training.** We conduct extensive empirical and theoretical analyses to validate our claim (Section 4 and Section 5). The main goal of Section 6 is not to propose new methods to achieve state-of-the-art performance, but to show that existing methods claiming to mitigate adversarial overfitting all implicitly avoid fitting hard adversarial examples. Furthermore, we show that preventing models from fitting hard adversarial examples can considerably improve the performance in the cases of adversarial finetuning and fast adversarial training.

---

### Author Response · Authors · 2021-11-20
**Revision Summary**

We thank all the reviewers for their constructive comments. These comments greatly help us improve our paper, making it clearer and rigorous. Below are the summary of our revision. The changes made in our paper are highlighted blue, including the main text and the appendix.

* **[Section 3 and Appendix D.1]** We clarify the properties of our proposed metric measuring the instance difficulty within a set. Given the clean instances and the adversarial perturbation type, the architecture and the training duration has little impact on that. In the last paragraph of Appendix D.1, which is added in the revision, we explain the motivation we define the instance difficulty as in Equation (1).

* **[Appendix D.3, Figure 7, 8]** We train the models as done in Figure 2(a) and Figure 2(b) for a longer time (600 epoch) until the loss values of the hard adversarial training instances saturate. The results are shown in Figure 7 and Figure 8 in Appendix D.3, respectively. Our conclusions in Section 4.1 and Section 4.2 remain valid.

* **[Appendix D.5, Figure 12]** We provided the learning curves of the methods studied in Appendix D.5.

* **[Appendix D.6, Figure 18, Figure 19]** We provide Figure 18 and Figure 19 in Appendix D.6 to demonstrate that the methods we use in Section 6 indeed avoid fitting hard adversarial training instances defined by our metric in Section 3.

---

### Decision · Program_Chairs · 2022-01-20

**Decision:**

Reject

**Comment:**

The paper proposes a metric to measure the difficulty of training examples. The main thesis is that hard training examples lead to bad test adversarial error. There are theoretical results on simple models establishing such claims. The paper also proposes a method to adaptively weight training examples to improve training which gives improvement for adversarial error.

The reviewers have raised a number of questions and the rebuttal period has been useful. In particularly, I agree with the reviewers that 'model-agnostic' is misleading in this context and the authors have agreed to remove this in the future. It is felt that more experiments, comparison to adversarial training, etc. is needed and I think the paper will need to go through a proper review process again before acceptance.